# A nucleation barrier spring-loads the CBM signalosome for binary activation

Alejandro Rodriguez Gama[1], Tayla Miller[1], Jeffrey J Lange[1], Jay R Unruh[1], Randal Halfmann[1,2]*

[1]Stowers Institute for Medical Research, Kansas City, United States; [2]Department of Molecular and Integrative Physiology, University of Kansas Medical Center, Kansas City, United States

**Abstract** Immune cells activate in binary, switch-like fashion via large protein assemblies known as signalosomes, but the molecular mechanism of the switch is not yet understood. Here, we employed an in-cell biophysical approach to dissect the assembly mechanism of the CARD-BCL10-MALT1 (CBM) signalosome, which governs nuclear transcription factor-κB activation in both innate and adaptive immunity. We found that the switch consists of a sequence-encoded and deeply conserved nucleation barrier to ordered polymerization by the adaptor protein BCL10. The particular structure of the BCL10 polymers did not matter for activity. Using optogenetic tools and single-cell transcriptional reporters, we discovered that endogenous BCL10 is functionally supersaturated even in unstimulated human cells, and this results in a predetermined response to stimulation upon nucleation by activated CARD multimers. Our findings may inform on the progressive nature of age-associated inflammation, and suggest that signalosome structure has evolved via selection for kinetic rather than equilibrium properties of the proteins.

## Editor's evaluation

Signalosomes are multi-protein complexes that transduce signals by sequentially assembling filamentous oligomers. Here, Rodriguez-Gama and colleague present their exciting finding in which the BCL10 adaptor in the CBM signalosome acts as an analogue-to-digital converter, resulting in binary activation of immune cells.

*For correspondence:
rhn@stowers.org

Competing interest: The authors declare that no competing interests exist.

## Introduction

Cells of both the innate and adaptive immune systems respond immediately and decisively to danger through the formation of large cytosolic protein complexes, known as signalosomes. Signalosomes function to coordinate the detection of pathogen- or danger-associated molecular patterns with protective transitions in cell state, such as programmed cell lysis or activation of nuclear transcription factor-κB (NF-κB) (*Kellogg et al., 2015*; *Liu et al., 2014*; *Matyszewski et al., 2018*; *Wu and Fuxreiter, 2016*). NF-κB directs the transcription of genes encoding pro-inflammatory cytokines and growth factors that are essential to both innate and adaptive immune responses, and its improper regulation contributes to cancer, chronic inflammation, and autoimmune diseases. Although NF-κB is known to respond in switch-like fashion to certain stimuli (*Kingeter et al., 2010*; *Muñoz et al., 2019*; *Tay et al., 2010*; *Figure 1A*), the thermodynamic basis of bistability has not been determined.

The CARD-BCL10-MALT1 (CBM) signalosome mediates NF-κB activation in lymphoid, myeloid, and certain nonimmune cell lineages. Named for its three protein constituents, the CBM signalosome is composed of: (1) a Caspase Activation and Recruitment Domain (CARD)-coiled coil (-CC) family member (either CARD9, 10, 11, or 14); (2) B-cell lymphoma 10 (BCL10); and (3) mucosa-associated

**eLife digest** The innate immune system is the body's first line of defence against pathogens. Although innate immune cells do not recognize specific disease-causing agents, they can detect extremely low levels of harmful organisms or substances. In response, they activate signals that lead to inflammation, which tells other cells that there is an infection. Innate immune cells are turned on in a switch-like fashion, becoming active very quickly after interacting with a pathogen. This is due to the action of signalosomes, large complexes made up of several proteins that clump together to form long chains that activate the cell.

But how do these large protein complexes assemble quick enough to create the switch-like activation observed in innate immune cells? To answer this question, Rodríguez Gama et al. focused on the CBM signalosome, which is involved in triggering inflammation through the activation of a protein called NF-kB.

First, Rodríguez Gama et al. used genetic tools to determine that activating the CBM signalosome drives a switch-like activation of NF-kB in cells. This means that individual cells in a population either become fully activated or not at all in response to minute amounts of harmful substances.

Once they had established this, Rodríguez Gama et al. wanted to know which protein in the CBM signalosome was responsible for the switch. They found that one of the proteins in the signalosome, called BCL10, has a 'nucleation barrier' encoded in its sequence. This means that it is very hard for BCL10 to start clumping together, but once it does, the clumps grow on their own. The nucleation barrier describes exactly how hard it is for these clumps to get started, and is determined by how disorganized the protein is.

When a pathogen 'stimulates' an immune cell, a tiny template is formed that lowers the nucleation barrier so that BCL10 can then aggregate itself together, leading to the switch-like behaviour observed. The nucleation barrier allows there to be more than enough BCL10 present in the cell at all times – ready to clump together at a moment's notice – and this permits the cell to detect very low levels of a pathogen.

Rodríguez Gama et al. then tested whether BCL10 from other animals also has a nucleation barrier. They found that this feature is conserved from cnidarians, such as corals or jellyfish, to mammals, including humans. This suggests that the use of nucleation barriers to regulate innate immune signalling has existed for a long time throughout evolution.

The work by Rodríguez Gama et al. broadens our understanding of how the innate immune system senses and responds to extremely low levels of pathogens. That BCL10 is always ready to clump together suggests it may be a driving force for chronic and age-associated inflammation. Additionally, the findings of Rodríguez Gama et al. also offer insights into how other signalosomes may become activated, and offer the possibility of new drugs aimed at modifying nucleation barriers.

lymphoid tissue lymphoma translocation protein 1 (MALT1) (*Gehring et al., 2018*; *Lu et al., 2019*). The CARD-CC proteins have tissue-specific expression that places CBM formation under the control of specific cell surface receptors. CARD11 expression in the lymphoid lineage controls B- and T-cell activation upon antigen recognition by B- and T-cell receptors (*Egawa et al., 2003*; *Wang et al., 2002*). CARD9 expression in the myeloid lineage links the antifungal response of monocytes to fungal carbohydrate detection by C-type lectin receptors (*Gross et al., 2006*; *Hsu et al., 2007*; *Strasser et al., 2012*). CARD10 and CARD14 are expressed in nonhematopoietic cells including intestinal and skin epithelia, respectively (*Ruland and Hartjes, 2019*). Multiple clinically significant mutations in CARD proteins have been found to compromise immune system homeostasis. For instance, mutations that relieve the normally autoinhibited state of CARD11 promote lymphoid cell proliferation leading to lymphoma (*Compagno et al., 2009*; *Lenz et al., 2008*). Several mutations in CARD9 cause familial hypersusceptibility to fungal infections (*Glocker et al., 2009*; *Lanternier et al., 2015*; *Lanternier et al., 2013*). CARD10 mutations are associated with risk of primary open angle glaucoma (*Zhou et al., 2016*), while CARD14 gain of function mutations cause psoriasis (*Fuchs-Telem et al., 2012*; *Jordan et al., 2012*). Once activated, CARD-CC proteins recruit the adaptor protein BCL10 and its binding partner, MALT1, resulting in the activation of downstream effectors that ultimately allow NF-κB to translocate to the nucleus and induce transcription of its target genes. As the integrators

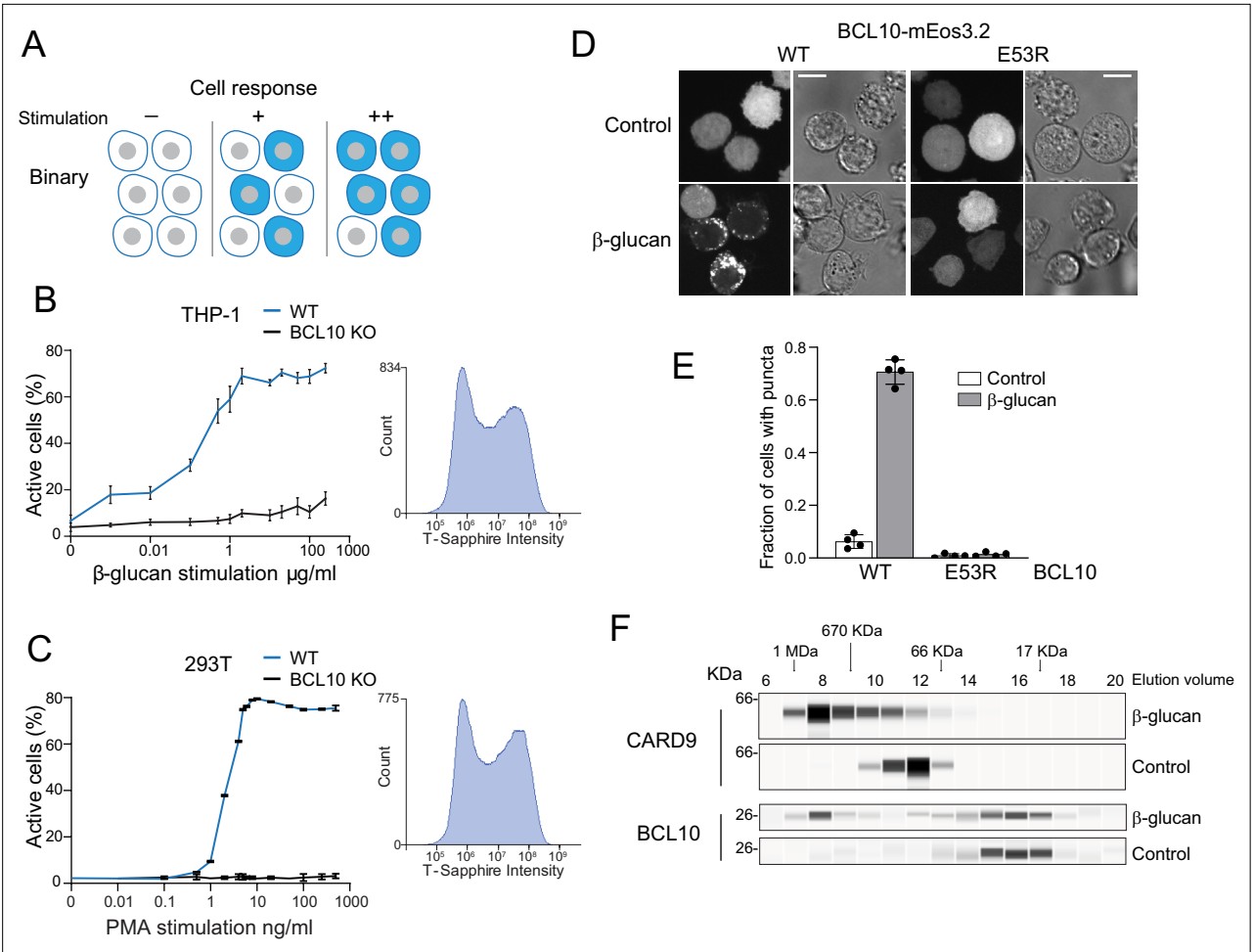

**Figure 1.** Assembly of the CARD-BCL10-MALT1 (CBM) signalosome drives all-or-none activation of nuclear transcription factor-$\kappa$B (NF-$\kappa$B). (**A**) Schematic diagram of binary activation in a population of cells. Higher doses of stimulation increase the probability, but not the magnitude, of activation by each cell. (**B**) β-Glucan stimulation of THP-1 WT and *BCL10*-KO monocytes transduced with an NF-$\kappa$B transcriptional reporter. Monocytes were stimulated for 24 hr then measured for T-Sapphire expression via flow cytometry. The graph shows the mean ± standard deviation (SD) fraction of cells positive for NF-$\kappa$B expression of three biological replicates. The inset shows the histogram of T-Sapphire expression indicating the distinct negative and positive populations of THP-1 WT cells upon stimulation with 10 µg/ml β-glucan. (**C**) HEK293 WT and *BCL10*-KO cells were transduced with an NF-$\kappa$B transcriptional reporter and stimulated with PMA for 24 hr then measured for T-Sapphire expression via flow cytometry. The inset shows the histogram of T-Sapphire expression indicating the distinct negative and positive populations of HEK293T WT cells upon stimulation with 5 ng/ml PMA. Shown are means ± SD of three biological replicates. (**D**) THP-1 *BCL10*-KO monocytes were transduced with a doxycycline-inducible BCL10-mEos3.2 construct. Cells were imaged for BCL10 expression 24 hr after Dox induction. Images show BCL10 puncta only in cells treated with β-glucan for 16 hr (see also *Figure 1—figure supplement 3*). In contrast, BCL10 E53R does not form puncta regardless of stimulation. Scale bar: 10 µm. (**E**) Quantification of the number of THP-1 cells with BCL10 puncta after β-glucan stimulation. Each dot represents the average of 4 independent experiments with more than 50 cells each. Shown are means ± SD. (**F**) Endogenous untagged CARD9 and BCL10 form high molecular weight species in THP-1 cells stimulated with β-glucan for 24 hr. After treatment, cells were lysed, and the protein extracts were resolved by size exclusion chromatography followed by capillary immunodetection.

The online version of this article includes the following source data and figure supplement(s) for figure 1:

**Source data 1.** Full image of protein immunodetection for CARD9 in lysates of THP-1 cells treated with β-glucan.

**Source data 2.** Full image of protein immunodetection for CARD9 in lysates of untreated THP-1 cells.

**Source data 3.** Full image of protein immunodetection for BCL10 in lysates of THP-1 cells treated with β-glucan.

**Source data 4.** Full image of protein immunodetection for BCL10 in lysates of untreated THP-1 cells.

**Figure supplement 1.** A single-cell nuclear transcription factor-$\kappa$B (NF-$\kappa$B) reporter reveals binary activation in both HEK293T and THP-1 cells.

**Figure supplement 2.** Generation and validation of reconstituted *BCL10* and *MALT1 KO* cell lines.

**Figure supplement 2—source data 1.** Full image of protein immunodetection for BCL10 and actin in lysates of THP-1 WT and BCL10-KO cells.

*Figure 1 continued on next page*

*Figure 1 continued*

**Figure supplement 2—source data 2.** Full image of protein immunodetection for BCL10, actin and MALT1 in lysates of HEK293T WT, BCL10-KO, and MALT1-KO cells.

**Figure supplement 2—source data 3.** Full image of protein immunodetection for BCL10 and actin in lysates of THP-1 WT cells and BCL10-KO cells reconstituted with BCL10-mEos3.2.

**Figure supplement 3.** BCL10 assembly underlies nuclear transcription factor-$\kappa$B (NF-$\kappa$B) activation in monocytes.

of cell type-specific CARD-CC signaling, BCL10 and MALT1 are expressed ubiquitously across cell types. Point mutations and translocations involving BCL10 and MALT1 cause immunodeficiencies, testicular cancer, and lymphomas (*Juilland and Thome, 2018*; *Kuper-Hommel et al., 2013*; *Ruland and Hartjes, 2019*).

Signalosome assembly involves homotypic polymerization by protein modules in the death domain (DD), Toll/interleukin-1 receptor (TIR), and RIP homotypic interaction motif (RHIM) families (*Nanson et al., 2019*; *Rodríguez Gama et al., 2021*; *Wu and Fuxreiter, 2016*). Some of these polymers, including those of BCL10, exhibit 'prion-like' self-templating activity in vitro or when overexpressed (*Cai et al., 2014*; *Franklin et al., 2014*; *Holliday et al., 2019*; *Hou et al., 2011*; *Kajava et al., 2014*; *Latty et al., 2018*; *Lu et al., 2014*; *Matyszewski et al., 2018*; *Mompeán et al., 2018*; *O'Carroll et al., 2020*; *Qiao et al., 2013*). Prion-like behavior is made possible by a structurally encoded kinetic barrier to de novo assembly, a process known as 'nucleation' (*Khan et al., 2018*; *Rodríguez Gama et al., 2021*; *Serio et al., 2000*). Nucleation involves a rare thermodynamic fluctuation whereby order spontaneously emerges from disorder (*Vekilov, 2012*). The nucleation barrier describes the rarity of that fluctuation in a given window of time and space.

For a signalosome protein whose activity is coupled to ordered polymerization, a sufficiently large nucleation barrier could allow cells to express the protein to concentrations that exceed its thermodynamic solubility limit, that is become supersaturated, *in anticipation of* pathogen exposure. Supersaturation would then provide a thermodynamic driving force for subsequently activating all of the excess molecules, thereby reducing the fraction of total molecules that need to be activated by directly interacting with – or 'sensing' – the pathogen. We posit that nucleation barriers could theoretically therefore allow immune systems to detect vanishingly small stimuli (perhaps even a single viral RNA; *Jiang, 2019*; *Zeng et al., 2010*). This mechanism would require (1) that the signalosome protein retains a nucleation barrier in the cellular milieu; (2) that its endogenous unstimulated intracellular concentration is supersaturating; and (3) that its polymerization can be triggered solely by the appearance of a nucleating particle in the cell, that is without the cell upregulating the protein or lowering its solubility in any way (e.g., via post-translational modifications or binding factors). These criteria have not yet been assessed for any signalosome.

In this study, we used a single-cell reporter of NF-κB activity to reveal that the CBM signalosome activates NF-κB in a binary fashion. We then used our recently developed biophysical approach, Distributed Amphifluoric FRET (DAmFRET; *Khan et al., 2018*; *Venkatesan et al., 2019*), to dissect the mechanism of CBM signalosome formation. This effort uncovered a structurally encoded nucleation barrier specifically in the adaptor protein, BCL10. We found that pre-existing CARD-CC protein multimers undergo a stimulus-dependent reorientation of their CARDs to create a template for BCL10 polymer nucleation. We further developed an optogenetic approach to nucleate BCL10 independently of upstream factors, revealing that BCL10 is indeed supersaturated even in resting cells. Finally, we showed that BCL10 activity does not depend on its polymer structure, implying that the structure is primarily a consequence of natural selection acting on the nucleation barrier. We found that the nucleation barrier is conserved from cnidaria to humans. Altogether, our work indicates that the preassembled CBM signalosome has evolved to store energy analogously to a spring-loaded mousetrap that allows cells to constitutively anticipate and respond to the slightest provocation.

## Results

### Assembly of the CBM signalosome drives all-or-none activation of NF-κB

To explore the link between signalosome nucleation and signaling kinetics, we first developed a single-cell reporter of NF-κB activity (*Figure 1—figure supplement 1A*). This transcriptional reporter contains four copies of the core NF-κB response element followed by the coding sequence of the fluorescent protein, T-Sapphire (*Wilson et al., 2013*). We used human embryonic kidney 293T cells and THP-1 monocytic cells to measure the activation of NF-κB with increasing concentrations of CBM signalosome stimulation. To activate the CBM signalosome in THP-1 monocytes, we stimulated CARD9 using the yeast cell wall component, β-glucan. To activate the CBM signalosome in HEK293T cells, we stimulated CARD10 (the only CARD-CC expressed in HEK293T cells) using the protein kinase C (PKC) activator, PMA (*Staal et al., 2021*). Using flow cytometry, we found that the percentage of T-Sapphire-positive cells increased in a dose-dependent manner for both HEK293T and THP-1 cells (*Figure 1B,C*). The distribution of T-Sapphire fluorescence was bimodal, with cells distributing between nonfluorescent and uniformly fluorescent populations. Increasing doses increased the fraction of cells in the fluorescent population, but did not influence the level of fluorescence within that population, even across multiple stimulation concentrations spanning several orders of magnitude (*Figure 1B,C* insets and *Figure 1—figure supplement 1B, C*). To ensure that activation was occurring solely through the CBM signalosome, we used CRISPR-Cas9 to disrupt exon 1 of *BCL10* in both HEK293T and THP-1 cells (*Figure 1—figure supplement 2A–C*). As expected, cells lacking BCL10 failed to respond to stimulation (*Figure 1B,C*).

This all-or-none switch from inactive to active NF-κB at the cellular level could be explained by a nucleation-limited transition of one or more signaling components to a stable assembly. We sought structural evidence of such a transition using size exclusion chromatography of lysates from THP-1 monocytic cell lines either with or without β-glucan stimulation, followed by immunodetection of both BCL10 and CARD9. We found that CARD9 (62 kDa) ran as an oligomer that, upon stimulation, shifted uniformly to a higher molecular weight (*Figure 1F*). BCL10 (26 kDa) also populated a large (>500 kDa) complex upon stimulation, although in this case, the size distribution was bimodal, with a fraction of the protein remaining monomeric. Because the protein does not populate intermediate-sized species, this result suggests an underlying phase transition.

To determine if the large multimers involve BCL10 polymerization, we knocked out *BCL10* in THP-1 cells and reconstituted it with either wild-type (WT) BCL10-mEos3.2 or polymerization-deficient mutant (E53R; *David et al., 2018*), expressed from a doxycycline-inducible promoter (*Figure 1—figure supplement 2D*). We then induced the expression of the WT and mutant proteins to similar levels, and then used microscopy to analyze the proteins' distributions in the presence or absence of β-glucan stimulation. In untreated cells, BCL10 fluorescence was entirely dispersed, consistent with the expected soluble form of the protein (*Figure 1D*). Conversely, in cells treated with β-glucan, the WT protein formed puncta in 70% of cells (*Figure 1E*), while the E53R mutant protein remained dispersed.

We also evaluated NF-κB activation in these reconstituted THP-1 *BCL10-KO* cells. To do so, we used an antibody against the NF-κB subunit, p65, to observe the protein's distribution between the nucleus and cytoplasm. We found that β-glucan-induced p65 nuclear translocation for cells reconstituted with WT, but not E53R mutant, BCL10-mEos3.2. Moreover, the degree of p65 nuclear translocation in the reconstituted cells resembled that of WT THP-1 cells treated with β-glucan (*Figure 1—figure supplement 3A, B*). Together, these results suggest that the CBM signalosome exerts binary control over NF-κB activation in monocytes.

### The BCL10 protein collective is a nucleation-mediated switch

We next sought to identify the specific protein(s) responsible for the switch. The assembly of certain innate immune signalosomes has been shown to involve prion-like self-templated polymerization of death fold domains (*Cai et al., 2014*; *Franklin et al., 2014*). CBM has a death fold domain in each of the three proteins – a CARD in CARD9/10/11/14, a CARD in BCL10, and a DD in MALT1 (*Figure 2A*). To determine whether one or more of these polymerizes in a switch-like fashion, we used DAmFRET to detect nucleation barriers for the full-length (FL) proteins (*Khan et al., 2018*; *Venkatesan et al., 2019*). In this method, a protein of interest is expressed as a genetic fusion to a photoconvertible fluorescent

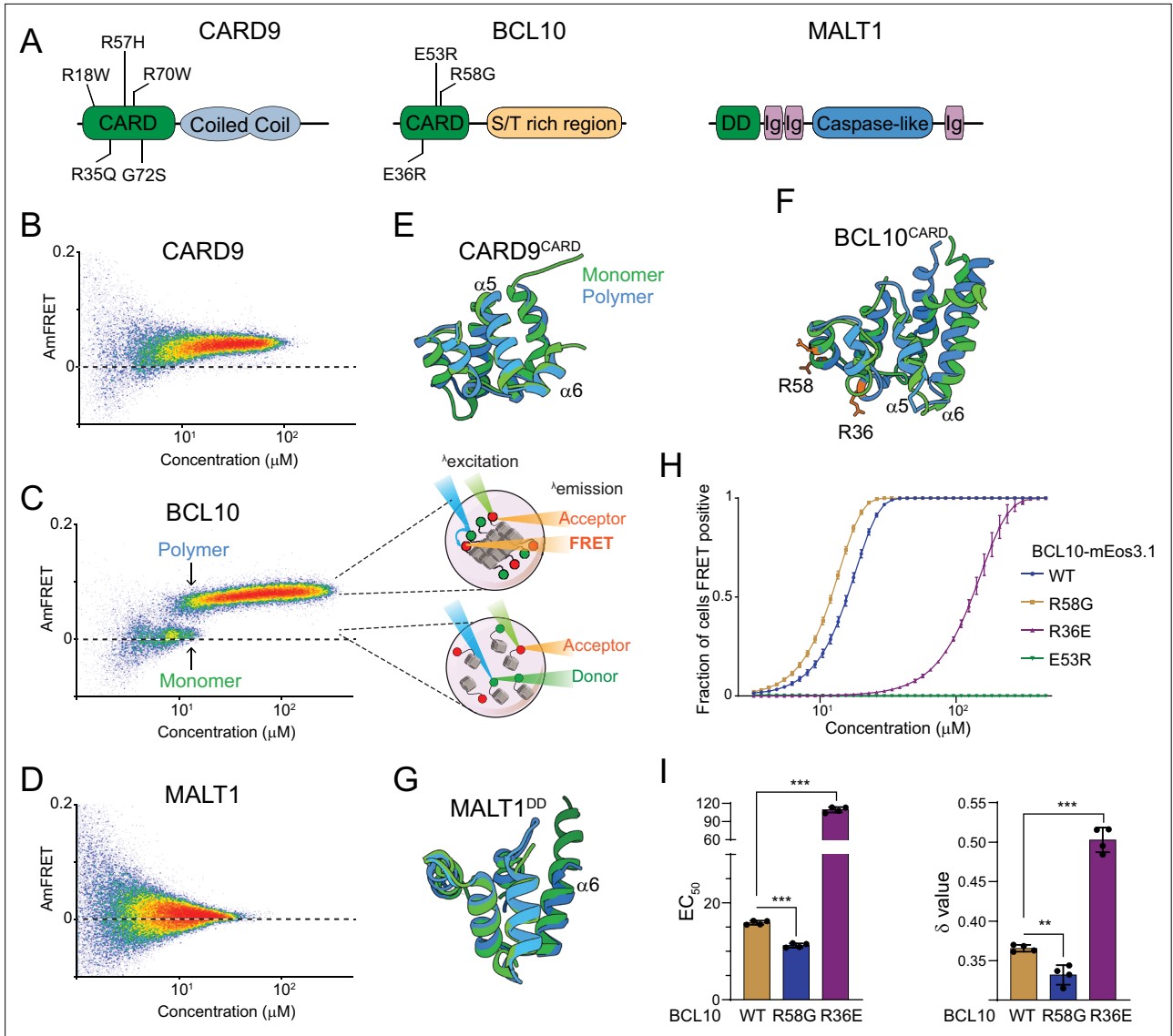

**Figure 2.** The adaptor protein BCL10 creates a nucleation-mediated switch. (**A**) Domain architecture of CARD-BCL10-MALT1 (CBM) proteins. Death fold domains are colored green. Mutations discussed in the text are indicated. (**B**) DAmFRET plot of CARD9 expressed in yeast cells, showing moderate AmFRET indicative of oligomer formation at all concentrations. (**C**) Left, DAmFRET plot of BCL10 expressed in yeast cells, showing a bimodal distribution between cells containing only monomers and cells containing polymers. The discontinuity and overlapping range of protein concentration for the two populations indicates that polymerization is rate limited by a nucleation barrier. Right, schema of cells in the two populations, showing how self-assembly of partially photoconverted mEos3.1-fused proteins causes a FRET signal. (**D**) DAmFRET plot of MALT1 expressed in yeast cells, showing no AmFRET, indicating a monomeric state of the protein at all concentrations. (**E**) Structural alignment of CARD9$^{CARD}$ monomers in its soluble (6E26, green) or polymerized (6N2P, blue) forms, showing no major conformational differences between them. (**F**) Structural alignment of BCL10$^{CARD}$ monomers in its soluble (2MB9, green) or polymerized (6BZE, blue) forms, showing multiple conformational differences that span a-helices 1, 2, 5, and 6. (**G**) Structural alignment of MALT1$^{DD}$ monomers in its soluble (2G7R, green) and polymerized (together with BCL10, 6GK2, blue) forms, showing no major conformational differences. (**H**) Weibull fits to DAmFRET plots of WT and mutant BCL10, showing how each mutation affects polymerization as quantified in (**I**). (**I**) EC$_{50}$ (left) and $\delta$ (right) values derived from the fits in (**H**). EC$_{50}$ describes the median concentration at which nucleation occurs. The slope parameter $\delta$ is a dimensionless proxy for the conformational free energy of nucleation. The R36E and E53R mutations, which lie in separate polymer interfaces, prevent the protein from polymerizing at low concentrations. The cancer-associated R58G mutation decreases the nucleation barrier, resulting in more frequent spontaneous nucleation relative to the WT protein. Shown are means ± standard deviation (SD) of three replicates. *t*-Test ***p < 0.001, **p < 0.005.

The online version of this article includes the following figure supplement(s) for figure 2:

**Figure supplement 1.** DAmFRET reveals the assembly mechanism of CARD-BCL10-MALT1 (CBM) components.

**Figure supplement 2.** BCL10 sequence conservation reveals selection for nucleation barrier.

protein, mEos3.1, from a genetic construct that varies in copy number between cells, so as to sample the broadest possible range of intracellular protein concentrations. An empirically optimized dose of blue light then photoconverts a fraction of the molecules in each cell to a FRET acceptor form, while unconverted molecules remain in the FRET donor form. Protein self-assembly increases FRET signal, which is evaluated as a function of the protein's concentration in each cell. We chose yeast cells as the reaction vessel because they contain neither death folds nor the associated signaling networks, which could otherwise obscure the exogenous proteins' sequence-encoded assembly properties. We found that neither CARD9 nor MALT1 formed self-templating polymers. The former formed low AmFRET multimers across the full range of concentrations (*Figure 2B*). These corresponded to the protein's accumulation into a single irregular punctum in each cell (*Figure 2—figure supplement 1A*). There was no discontinuity in DAmFRET and hence no observable nucleation barrier to the formation of these puncta at the cellular level. MALT1 remained completely monomeric (*Figure 3D*, and *Figure 2—figure supplement 1C*). BCL10, in contrast, produced a clear bimodal distribution wherein cells partitioned between a no-AmFRET population and a high-AmFRET population, respectively, comprised of cells containing only monomeric BCL10 or highly polymerized BCL10 (*Figure 2C* and *Figure 2—figure supplement 1B*). The two populations overlapped in their expression levels, indicating that BCL10 can accumulate to supersaturating concentrations while remaining monomeric, but then polymerizes following a stochastic nucleation event. The paucity of cells between the two populations, that is with moderate levels of AmFRET, suggests that BCL10 polymerizes rapidly to a new steady state following a single nucleation event in each cell (*Khan et al., 2018*; *Posey et al., 2021*).

We next evaluated nucleation barriers of the isolated death fold domains of CARD9, BCL10, and MALT1. MALT1$^{DD}$ remained monomeric at all concentrations (*Figure 2—figure supplement 1F*). CARD9$^{CARD}$ was monomeric up to a threshold concentration of approximately 100 μM, above which it readily polymerized with only a slight discontinuity (*Figure 2—figure supplement 1D*). In contrast, BCL10$^{CARD}$ switched from monomer to polymer with a large discontinuity (*Figure 2—figure supplement 1E*) resembling that of FL BCL10, suggesting that the large nucleation barrier of the FL protein, and by extension the switch-like assembly of the CBM assembly, can be attributed to the CARD of BCL10.

The structural basis of nucleation barriers that allow for certain death fold domains to supersaturate in cells is not completely understood. We reasoned that, as for amyloid-based prions, it may result from a conformational change required for subunit polymerization. To investigate this possibility, we took advantage of previously solved structures for all three of the CBM death fold domains. By computationally superposing the structures of the monomer and polymer forms for each domain, and calculating the root mean square deviation (RMSD) between backbone C-alpha atoms, we found that BCL10$^{CARD}$ undergoes a greater conformational change to polymerize than does either CARD9$^{CARD}$ or MALT1$^{DD}$ (*Figure 2E–G*; RMSD 3.581, 1.655, and 0.956 Å, respectively). Although more sophisticated analyses are required to relate these differences to nucleation barriers, they are consistent with the nucleation barrier of BCL10$^{CARD}$ resulting from a particularly unfavorable conformational fluctuation.

If the monomeric fold of BCL10 functions to restrict nucleation, then mutations that relax that fold can be expected to pathogenetically activate the CBM signalosome. The R58G mutation of BCL10 was first described in a germ cell line tumor and shown to hyperactivate NF-κB (*Willis et al., 1999*). We tested via DAmFRET if this can be attributed to a reduction in the sequence-encoded nucleation barrier. The EC$_{50}$ and $\delta$ (delta) statistics obtained by fitting DAmFRET to a Weibull function serve as crude proxies for the inter- and intramolecular free energies of nucleation, respectively (*Figure 2—figure supplement 1G*). More specifically, they describe the median concentration at which nucleation occurs, and the independence of nucleation on concentration, respectively, where conformationally limited nucleation has higher $\delta$ values (*Khan et al., 2018*). Indeed, relative to WT, R58G reduced both $\delta$ and EC$_{50}$ (*Figure 2H,I*). In contrast, mutation R36E, which disrupts polymer interface IIa (*David et al., 2018*), increased both $\delta$ and EC$_{50}$ (*Figure 2I*). Consistent with previous observations, mutant E53R, which disrupts polymer interface IIIb (*David et al., 2018*), blocked nucleation at all tested concentrations (*Figure 2H*). We next investigated the evolutionary conservation of R58. We found via multiple sequence alignment that position 58 is an arginine in mammals but a glutamine in lamprey, cartilaginous fish, and amphibians (*Figure 2—figure supplement 2A*). We therefore evaluated the nucleation barrier when R58 is substituted with Gln or with a hydrophobic residue of comparable size (Leu). As expected, R58Q resembled WT, whereas R58L lowered the nucleation barrier (*Figure 2—figure*

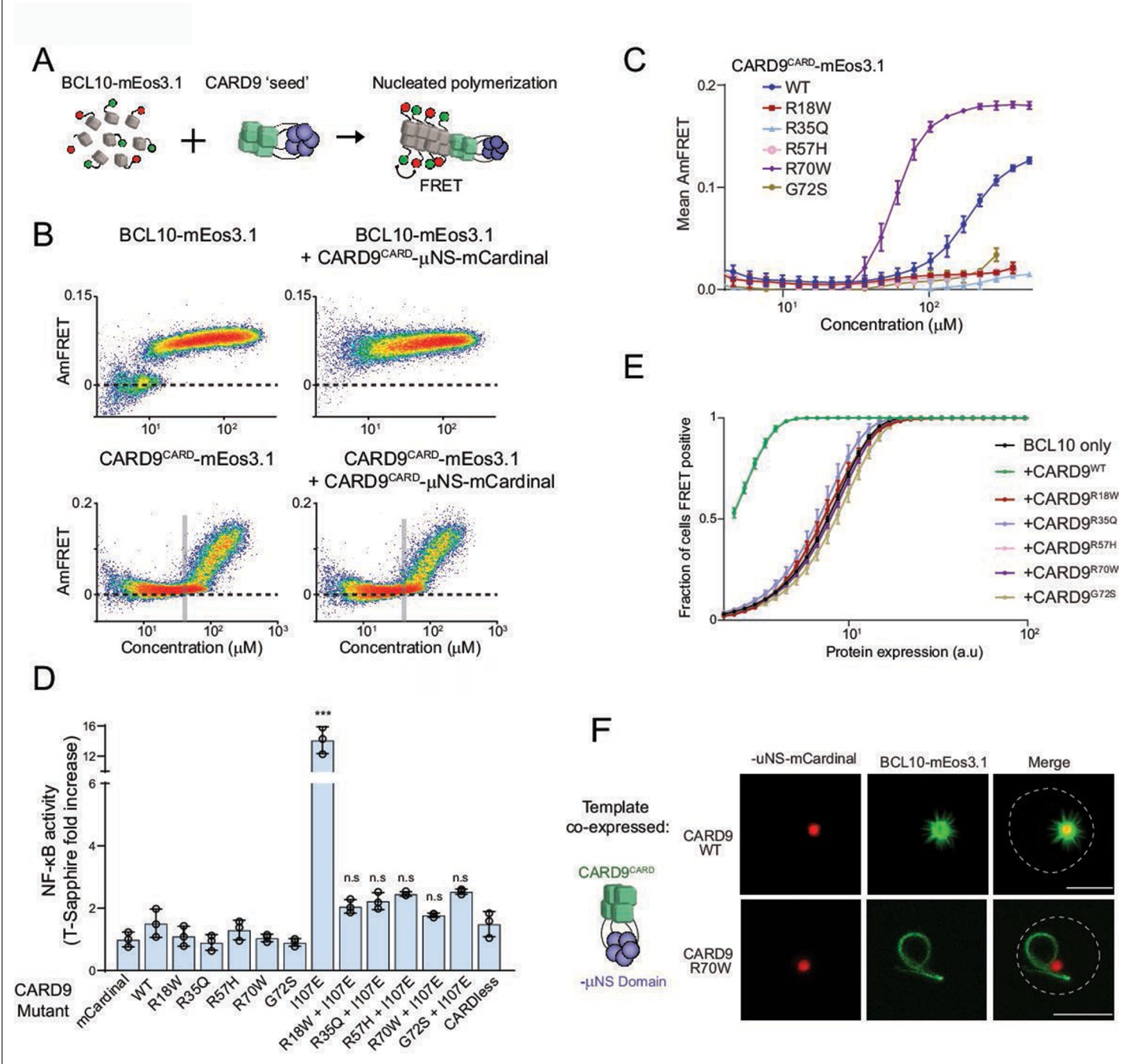

**Figure 3.** Pathogenic mutations disrupt BCL10 template formation within CARD-CC multimers. (**A**) Schematic of the DAmFRET experiment used to test for nucleating interactions between CARD-BCL10-MALT1 (CBM) components. Artificial 'seeds' are created by expressing the putative nucleating protein as a fusion to the homo-oligomerizing module, μNS (purple). (**B**) DAmFRET plots of BCL10 (top) or CARD9$^{CARD}$ (bottom) in the absence (left) or presence (right) of CARD9$^{CARD}$ seeds expressed in trans. For BCL10, the presence of seeds shifted all low-AmFRET cells to high-AmFRET, indicating that the protein had been supersaturated and is nucleated by CARD9$^{CARD}$. In contrast, the seeds had no effect on CARD9$^{CARD}$ DAmFRET, suggesting that CARD9$^{CARD}$ only polymerizes at high concentrations and does not become supersaturated. The vertical line denotes the phase boundary for CARD9$^{CARD}$ polymerization. (**C**) Graph of spline fits of AmFRET values for CARD9$^{CARD}$ mutants. All of the pathogenic mutations within the CARD domain disrupted polymer formation except for R70W, which stabilized polymers. (**D**) Effect of CARD9 mutations in the full-length protein context with autorepression eliminated by the I107E mutation. 293T cells transduced with the nuclear transcription factor-κB (NF-κB) reporter were transfected with the indicated CARD9-mCardinal constructs and analyzed for T-Sapphire expression at 48-hr post-transfection. T-Sapphire expression was normalized to that of cells transfected with plasmid expressing mCardinal alone. Shown are means ± standard deviation (SD). Analysis of variance (ANOVA) with post hoc Tukey test ***p < 0.0001. (**E**) Weibull fits to DAmFRET plots of BCL10 coexpressed with mutant CARD9 seeds, showing that only WT CARD9 nucleates BCL10

*Figure 3 continued on next page*

*Figure 3 continued*

while none of the pathogenic mutants do so. (**F**) Images of yeast cells expressing the respective CARD9 seeds (red punctum) and BCL10-mEos3.1 (green). CARD9 WT seeds nucleate BCL10 polymers, resulting in a starburst-like structure. Conversely, CARD9-mutant R70W does not nucleate BCL10; BCL10 instead forms a single long filament that is detached from the 'seed', indicating that it polymerizes spontaneously in these cells. Scale bar: 5 µm.

The online version of this article includes the following figure supplement(s) for figure 3:

**Figure supplement 1.** Dissection of CARD9 multimerization reveals a critical role of its coiled-coil region.

**Figure supplement 2.** CARD9$^{CARD}$ polymers and other CARD-CC proteins can nucleate BCL10.

**Figure supplement 3.** Pathogenic mutations in CARD9$^{CARD}$ reveal the structural basis of functional CARD-BCL10-MALT1 (CBM) signalosome formation.

*supplement 2B, C*). These data suggest that Arg or Gln residues at position 58 functionally oppose nucleation and thereby prevent precocious activation of BCL10 in the germ cell lineage. Altogether, we conclude that the native monomeric structure of BCL10 creates a physiological nucleation barrier that allows for switch-like self-assembly and all-or-none NF-κB activation.

## Pathogenic mutations disrupt BCL10 template formation within CARD-CC multimers

CARD-CC proteins form oligomers that promote BCL10 polymerization (*David et al., 2018*; *Holliday et al., 2019*). Somewhat paradoxically, however, CARD-CC proteins are believed to reside in an oligomeric state even in the absence of stimulation. The active configuration is prevented from forming within these oligomers due to an inhibitory interaction between the CARD and the adjacent coiled-coil region (*Holliday et al., 2019*; *Sommer et al., 2005*). That interaction is released by post-translational modifications induced by pathway activation, leading to CBM assembly. The very low affinity of CARD9$^{CARD}$ for itself (*Figure 2—figure supplement 1D* and *Holliday et al., 2019*) suggests that nucleation is unlikely to occur at physiological concentrations without assistance from coiled coil-mediated multimerization. This in turn necessitates that the coiled-coil interactions support a stoichiometry higher than the dimer that we had previously observed for CARD9$^{1-142}$ (*Holliday et al., 2019*), as a dimer has too few interfaces to template the four-start helical polymer of BCL10 (*Qiao et al., 2013*). Our observation that FL CARD9 forms a punctum at all expression levels, whereas the CARD itself forms either monomers (at low expression) or polymers (at high expression), is consistent with an essential role of non-CARD interactions in both autoinhibition and higher-order assembly.

To further dissect the relationship between multimerization, autoinhibition, and nucleation, we first used DAmFRET to assess CARD9 multimerization in the absence of its CARD. Remarkably, this truncated protein aggregated just as robustly as FL (*Figure 3—figure supplement 1B, C*), suggesting that coiled-coil interactions are predominantly if not entirely responsible for its multimerization in the inactive state. Next, we progressively truncated the non-CARD region of the protein (*Figure 3—figure supplement 1A*). We found that even the shortest variant tested (1–142), retaining only 44 residues of additional sequence beyond the CARD, multimerized at all concentrations (*Figure 3—figure supplement 1B, C*). Unlike longer versions of the protein, however, this variant populated a lower AmFRET state and exclusively soluble oligomers, consistent with the reduced valency of the truncated coiled-coil region (*Figure 3—figure supplement 1D*). Importantly, none of these constructs formed the higher AmFRET state or fibrillar puncta that would be indicative of polymerization. This tight relationship between multimerization and inhibition can be rationalized by the previously solved structure of the autoinhibited dimer formed by the short variant, wherein each CARD packs against the broad face of the immediately adjacent dimeric coiled-coil, making extensive contacts with *both* helices (*Holliday et al., 2019*).

We had previously demonstrated that point mutations in the coiled-coil-CARD interface release autoinhibition (*Holliday et al., 2019*). We therefore asked if the best characterized such mutation, I107E, has any effect on multimerization. Relative to WT, the mutant I107E achieved higher AmFRET at low expression and rendered the punctum less spherical (*Figure 3—figure supplement 1E, F*). Together with the prior demonstration of functional activity, the present findings suggest that multimerization itself is not inhibitory to nucleation and that once autoinhibition is released, CARD–CARD interactions stabilize the aggregated state. We therefore speculate that pre-multimerization driven by the multivalent coiled-coil expedites the response to stimulation, perhaps by allowing stimulus-dependent post-translational modifications (*Zhong et al., 2018*) to occur cooperatively on multiple

CARD9 subunits in close proximity. This would reduce the entropic cost of CARD9 activation relative to the subunits oligomerizing de novo.

We noticed that high expression levels of CARD9$^{CARD}$-mCardinal sufficed to nucleate BCL10-mEos3.1, whereas mCardinal itself had no such activity (*Figure 3—figure supplement 2A,C*), suggesting that polymers of CARD9$^{CARD}$ itself can nucleate BCL10. To test if a high local concentration suffices for CARD9$^{CARD}$ subunits to organize into BCL10 polymer nuclei, we substituted the coiled-coil region with an inert and well-characterized homomultimeric domain, μNS (*Schmitz et al., 2009*). This domain forms stable condensates that drive its fusion partner to very high local concentration (*Figure 3A*) without affecting global expression levels (*Figure 3—figure supplement 2D*), and thereby eliminates intermolecular entropic contributions to nucleation barriers even at low expression (*Kandola et al., 2021*). Expressing the CARD9$^{CARD}$-μNS-mCardinal fusion should trigger the high-AmFRET state of any yeast cell expressing either CARD9$^{CARD}$- or BCL10-mEos3.1 above their respective solubility thresholds. As hypothesized, CARD9$^{CARD}$-μNS-mCardinal shifted all BCL10-mEos3.1 yeast cells to the high-FRET population (*Figure 3B*), confirming that CARD9$^{CARD}$ indeed forms a polymer seed when condensed. Moreover, the data indicate that in the absence of a template, BCL10 is supersaturated even at the lowest levels of expression (nanomolar). In striking contrast, CARD9$^{CARD}$-μNS-mCardinal had no effect on the DAmFRET profile of CARD9$^{CARD}$-mEos3.1, suggesting that CARD9$^{CARD}$ instead forms labile polymers with no observable nucleation barrier at the cellular level (*Figure 3B*).

We next asked if the other CARD-CC family members – CARD10, 11, and 14 – function in the same fashion. These proteins have the same domain architecture as CARD9: a CARD followed by coiled-coil domains. We performed DAmFRET on their CARD and found that all polymerized comparably to that of CARD9 (*Figure 3—figure supplement 2E*), that is only at high concentrations (the most stable polymers were formed by CARD10, but even then only above approximately 40 μM), and with negligible nucleation barriers. We then asked if they too, when artificially condensed by fusion to μNS-mCardinal, suffice to nucleate BCL10. Indeed all exhibited this activity (*Figure 3—figure supplement 2F*). These findings suggest a shared mechanism of action by all four CARD-CC proteins.

Several mutations in CARD9$^{CARD}$ cause susceptibility to fungal infections in humans (*Figure 2A*). We noted that these residues are conserved within the CARD-CC proteins (*Figure 3—figure supplement 3A*). To determine their mode(s) of action, we first asked if the mutants can form the polymer structure using DAmFRET and fluorescence microscopy. Whereas WT CARD9$^{CARD}$ assembled into polymers at high concentration, the pathogenic mutants R18W, R35Q, R57H, and G72S failed to do so (*Figure 3C* and *Figure 3—figure supplement 3B, C*). Remarkably, however, the R70W mutant not only retained the ability to polymerize, but did so at lower concentrations than WT (*Figure 3C* and *Figure 3—figure supplement 3B, C*), implying that its defect instead lies downstream of multimerization. We then assessed the ability of the CARD9 pathogenic mutations to nucleate BCL10, by expressing the CARD9$^{CARD}$ mutants as fusions to μNS in yeast cells simultaneously expressing BCL10-mEos3.1. As expected CARD9$^{CARD}$ WT nucleated BCL10-mEos3.1 at all expression levels. Using confocal fluorescence microscopy, we observed the BCL10-mEos3.1 filaments emanating directly from the CARD9$^{CARD}$-μNS-mCardinal puncta. In contrast, none of the pathogenic mutants induced BCL10-mEos3.1 nucleation nor observable filaments (*Figure 3E,F*). Collectively, these data reveal that R18W, R35Q, R57H, and G72S are structurally incompatible with the polymeric configuration, whereas R70W specifically disrupts the interaction of CARD9 polymers with BCL10 at interface IIb (*Figure 3—figure supplement 3E*).

To investigate downstream consequences of the CARD mutations, and in the context of the FL protein, we next transiently transfected our HEK293T NF-κB reporter cells with FL versions of CARD9, all harboring the I107E mutation to eliminate autoinhibition. We first confirmed that CARD9 I107E strongly activated NF-κB, and that a construct lacking the CARD did not (*Figure 3D*). We next introduced each of the pathogenic mutants into this construct and found that every one of them eliminated the protein's ability to activate NF-κB (*Figure 3D*). All variants expressed to similar levels as WT (*Figure 3—figure supplement 3D*). Altogether, these results suggest that CARD9 and its orthologs function to nucleate BCL10 by forming a polymeric structure stabilized by multivalent coiled-coil interactions, and that disrupting that activity compromises innate immunity.

## The BCL10 nucleation barrier causes binary activation of NF-κB

We next asked if switch-like BCL10 polymerization results in binary activation of NF-κB in human cells. To do so, we monitored NF-κB activation with respect to polymerization by CBM signalosome components. We transfected constructs of proteins fused to mEos3.2 into HEK293T cells containing the transcriptional reporter (T-Sapphire) of NF-κB activation (*Figure 4A*). Using DAmFRET, we confirmed that all of the proteins behaved the same way in HEK293T cells as they did in yeast. Specifically, FL CARD9 formed irregular higher-order assemblies at all concentrations (*Figure 4—figure supplement 1A,C*); CARD9^CARD polymerized only above a high threshold expression level; MALT1 was entirely monomeric (*Figure 4—figure supplement 1A*); and BCL10 became supersaturated with respect to a polymerized form (*Figure 4B*). Neither CARD9 nor MALT1 expression activated NF-κB (*Figure 4—figure supplement 1A,B*). BCL10 expression, in contrast, robustly activated NF-κB (*Figure 4C*). Importantly, this activation only occurred in cells that contained BCL10 polymers, and not monomers, irrespective of the expression level of BCL10 (*Figure 4C*, *Figure 4—figure supplement 1D, E*). The frequency of activation, but not the level of activation within individual cells, was reduced or eliminated for the polymer-inhibiting mutants R36E and E53R, respectively (*Figure 4—figure supplement 1D, E*). Mutant R58G increased the fraction of cells with activated NF-κB (*Figure 4—figure supplement 2A–C*), as expected from its reduced nucleation barrier. Importantly, the intensity of NF-κB response at the cellular level did not differ between the mutants (*Figure 4—figure supplement 1E* and *Figure 4—figure supplement 2A–C*), suggesting a specific effect on nucleation and not downstream signaling. To determine if activation was specific to BCL10 polymers, we also performed the experiment with ASC, the death fold-containing adaptor protein of a different signalosome (the inflammasome). As expected (*Cai et al., 2014*), ASC formed prion-like polymers (*Figure 4B*), but did not activate the NF-κB reporter (*Figure 4C*). This experiment suggests that polymers of BCL10, specifically, and independently of upstream components or physiological stimuli, suffice to activate downstream components of the pathway, and that an intrinsic nucleation barrier to their formation causes activation to occur in a binary fashion.

## Endogenous BCL10 is constitutively supersaturated

While the results obtained thus far indicate that BCL10 *can* function as a switch via nucleation-limited polymerization, we next asked whether it indeed does so even at endogenous levels of expression, that is in the absence of overexpression to promote BCL10 assembly. This distinction is very important because it determines if the nucleation barrier is responsible for preventing NF-κB activation in the absence of stimulation. To determine if the protein is *constitutively* supersaturated, we designed the experiment to eliminate any upregulation of *BCL10* transcription or translation that might occur upon stimulation (*Yan et al., 2008*). Specifically, we reconstituted our BCL10-deficient HEK293T cells with BCL10-mScarlet expressed from a doxycycline-inducible promoter. After isolating a single clone with uniform expression, we performed a doxycycline titration and used capillary protein immuno-detection to compare the expression level of reconstituted BCL10-mScarlet to that of endogenous BCL10 in unstimulated HEK293T. We found that 1 μg/ml doxycycline-induced BCL10-mScarlet to approximately the same level as endogenous BCL10 (*Figure 4—figure supplement 3A–C*). We next compared BCL10 expression levels in HEK293T cells to that of THP-1 cells and primary human fibroblasts, and found comparable expression levels across all three cell lines (*Figure 4—figure supplement 3D, E*), consistent with the known ubiquity of the CBM signalosome. Finally, we analyzed BCL10 expression at the single-cell level using immunofluorescence and flow cytometry, which revealed that the reconstituted expression of BCL10-mScarlet has the same median intensity of BCL10 antibody binding as that of endogenous BCL10 in unmodified HEK293T cells (*Figure 4—figure supplement 3F*).

This cell line enables us to monitor NF-κB activity with respect to the assembly state of BCL10 at endogenous unstimulated levels of expression. To achieve this, we introduced a fluorescently tagged NF-κB subunit, EYFP-p65, and performed time-lapse fluorescence microscopy following the addition of PMA. Prior to stimulation, both BCL10 and p65 were diffusely distributed throughout the cytoplasm. BCL10 became punctate within 50 min after PMA addition, and the puncta could be observed to elongate with time (*Figure 4D,E* and *Video 1*). Concomitantly, p65 was observed to translocate to the nucleus (*Figure 4E*). This result suggests that BCL10 is indeed supersaturated prior to stimulation.

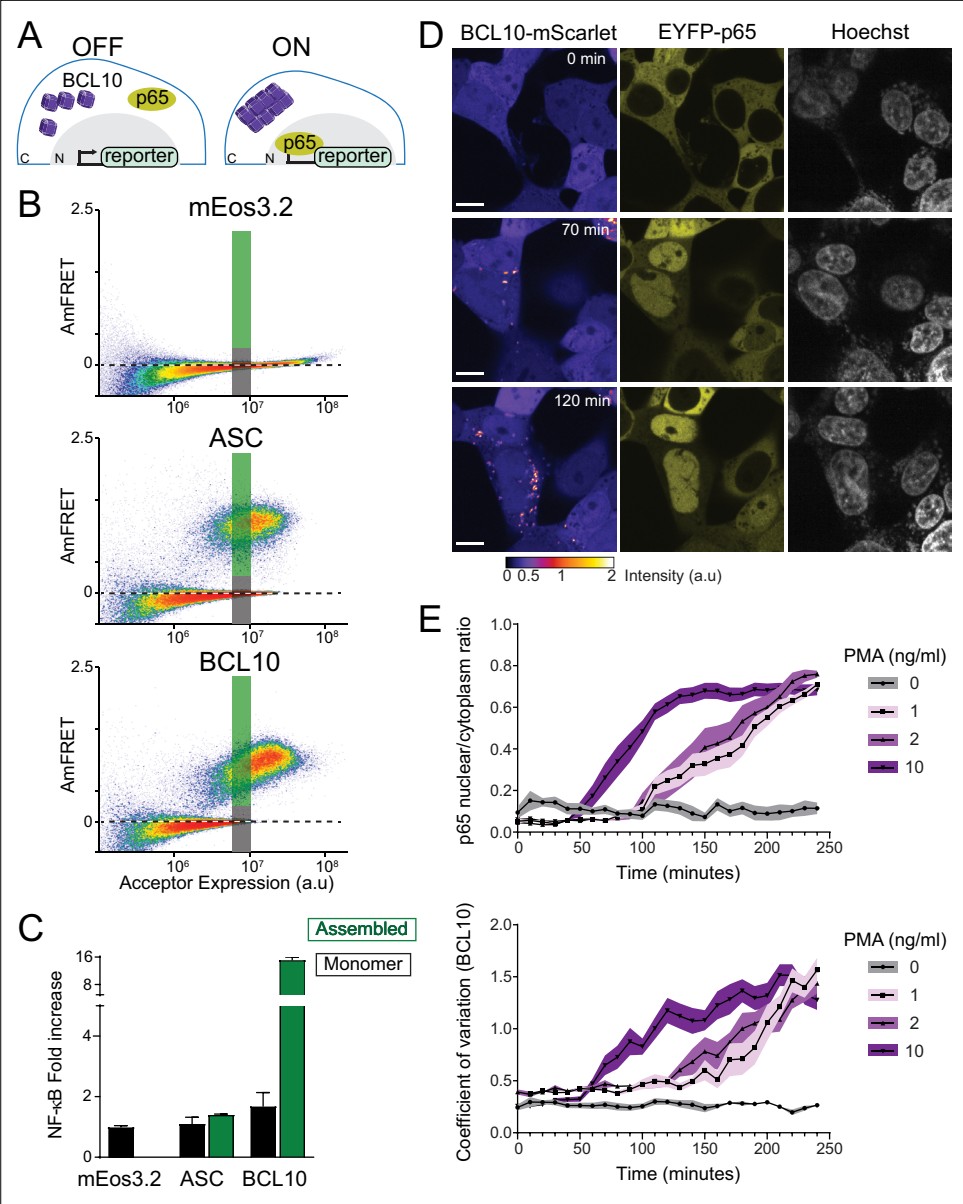

**Figure 4.** The BCL10 nucleation barrier causes binary activation of nuclear transcription factor-$\kappa$B (NF-$\kappa$B). (**A**) Schematic of experiments to determine if BCL10 polymerization activates NF-$\kappa$B in 293T cells. In the OFF state, BCL10 is dispersed in the cell and the p65 NF-$\kappa$B subunit is localized to the cytosol. In the ON state, BCL10 is polymerized and p65 relocates to the nucleus, where it induces the transcription of NF-$\kappa$B response genes. (**B**) DAmFRET plots of the indicated proteins expressed in 293T cells containing the NF-$\kappa$B reporter, 48 hr after transfection. The reporter mEos3.2 remains in the monomeric state and does not activate NF-$\kappa$B. The gray and green boxes designate the regions gated for cells containing either monomer or polymerized protein, respectively, and analyzed for T-Sapphire expression in (**C**). (**C**) Quantification of T-Sapphire expression in the gated areas of the DAmFRET plots shown in (**B**). Shown are means ± standard deviation (SD). Only cells that contained BCL10 polymers, but not cells containing the same concentration of monomeric BCL10 nor cells containing ASC polymers, activated NF-$\kappa$B. (**D**) Time-lapse microscopy of 293T *BCL10*-KO cells reconstituted with *BCL10*-mScarlet and EYFP-p65. Cells were stimulated with 10 ng/ml PMA and imaged every 10 min. BCL10 and p65 are distributed throughout the cytosol at 0 min, but respectively relocalize to cytosolic puncta and the nucleus over the course of the experiment. Scale bar: 10 µm. (**E**) Top, ratios of EYFP fluorescence intensities in the nucleus versus cytosol; and bottom, values of the coefficient of variation of mScarlet pixel intensities, over the time course of cells stimulated with PMA at the indicated concentrations. BCL10 polymerization produced visible puncta within approximately

*Figure 4 continued on next page*

*Figure 4 continued*

50 min of PMA stimulation. More than 40 cells were tracked and analyzed for each treatment condition. Shown are means ± SD.

The online version of this article includes the following source data and figure supplement(s) for figure 4:

**Figure supplement 1.** DAmFRET coupled with nuclear transcription factor-κB (NF-κB) single-cell reporter cell line reveals CARD-BCL10-MALT1 (CBM) signalosome components contribution to binary activation.

**Figure supplement 2.** BCL10 intrinsic nucleation barrier dictates binary activation of nuclear transcription factor-κB (NF-κB).

**Figure supplement 3.** Reconstitution of HEK293T *BCL10-KO* cells allows for testing BCL10 endogenous supersaturation.

**Figure supplement 3—source data 1.** Full image of protein immunodetection for BCL10 and actin in lysates of HEK293T WT cells and BCL10-KO cells reconstituted with BCL10-mScarlet.

**Figure supplement 3—source data 2.** Full image of protein immunodetection for BCL10 and actin in lysates of HEK293T WT, THP-1, fibroblasts, and HEK293T BCL10-KOcells.

**Figure supplement 4.** BCL10 supersaturation is necessary and sufficient to activate nuclear transcription factor-κB (NF-κB).

---

Two key predictions of our hypothesis are that (1) BCL10 will be unable to respond to stimulation when its concentration falls below a threshold (corresponding to its solubility limit), and (2) it will activate spontaneously even in the absence of stimulation when its concentration far exceeds that threshold. To test the first prediction, we induced BCL10-mScarlet to subphysiological levels using 0.05 µg/ml doxycycline, stimulated the cells with PMA, and then used confocal microscopy to quantify the nucleocytoplasmic distribution of EYFP-p65 as a function of BCL10-mScarlet intensity. As expected, EYFP-p65 failed to translocate to the nucleus in cells with BCL10-mScarlet intensity below an apparent threshold (*Figure 4—figure supplement 4A, B*). To test the second prediction, we induced BCL10-mScarlet to superphysiological levels by administering 1 µg/ml doxycycline for extended durations, while withholding PMA. By 24 hr of doxycycline induction, a small fraction of cells (1.1%) had spontaneously acquired BCL10 puncta. This fraction increased to 5.1% at 48 hr and 7.8% at 72 hr (*Figure 4—figure supplement 4C, D*). As expected, the cells containing BCL10 puncta also showed EYFP-p65 translocation (*Figure 4—figure supplement 4C*). We conclude that BCL10 supersaturation is necessary to activate NF-κB, while deep supersaturation is sufficient.

As a final test that BCL10 is physiologically supersaturated, we next devised an optogenetic approach to nucleate BCL10 independently of overexpression and independently of potential orthogonal cellular response to stimulation, such as post-translational modifications or changes in the binding of other proteins to BCL10 that could lower its solubility limit. Cry2clust is a variant of a plant photoreceptor that reversibly homo-oligomerizes in response to blue light excitation (*Park et al., 2017*). We expressed CARD9CARD as a fusion to miRFP670-Cry2clust (CARD-9CARD-Cry2) in the HEK293T *BCL10*-KO cells with BCL10-mScarlet reconstituted to endogenous levels (*Figure 5A*), and used a brief (1 s) pulse of 488 nm laser light to activate Cry2. Prior to blue light exposure, both CARD9CARD-Cry2 and BCL10-mScarlet were diffusely distributed throughout the cells (*Figure 5B* and *Video 2*). Following the pulse, however, both proteins assembled into puncta. By measuring the kinetics of CARD-9CARD and BCL10 assembly (as the coefficient of variation in pixel intensity over time), we found that CARD9CARD clustering peaked at approximately 5-min post-excitation, and then those clusters disassembled to background levels by approximately 20 min (*Figure 5C*). Within 6 min following the pulse, BCL10 began to form filamentous puncta that initially colocalized with those of CARD9CARD (*Figure 5B*). Remarkably, the puncta continued to elongate even after the CARD9CARD clusters had dissolved (*Figure 5B*

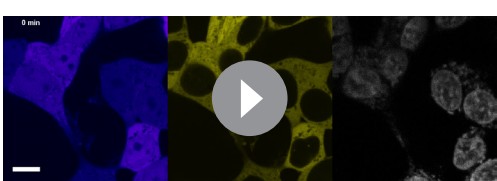

**Video 1.** BCL10 nucleation instructs p65 nuclear translocation. This video shows HEK293T *BCL10-KO* cells reconstituted with BCL10-mScarlet expressing EYFP-p65. Upon addition of PMA, cells acquire BCL10 puncta (left panel) and p65 nuclear translocation (middle panel). Right panel, Hoechst staining. Scale bar: 10 µm.
https://elifesciences.org/articles/79826/figures#video1

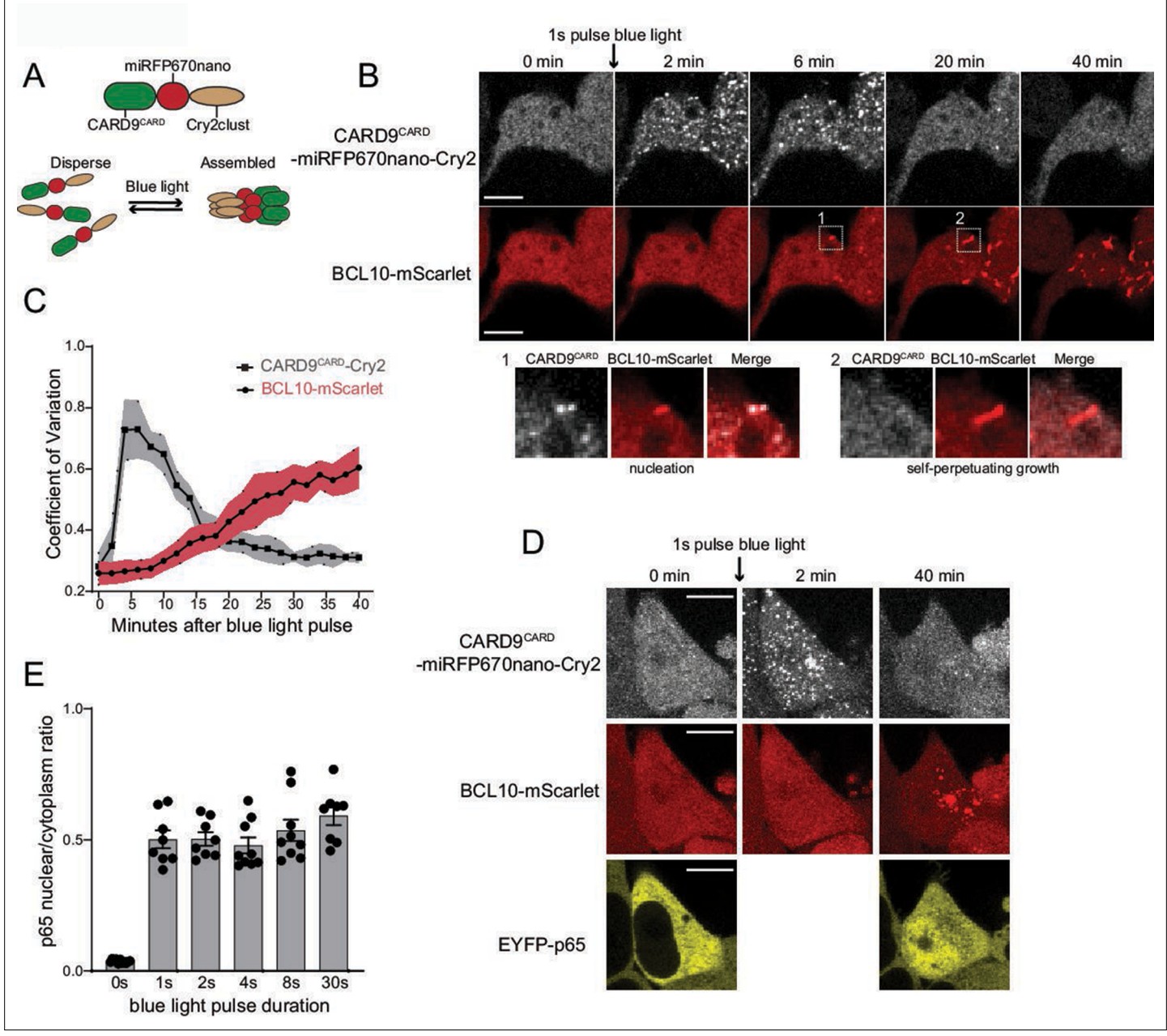

**Figure 5.** Endogenous BCL10 is constitutively supersaturated. (**A**) Schematic of the optogenetic nucleation experiment. CARD9[CARD] is expressed as a fusion to the near-infrared fluorescent protein miRFP670nano and the light-controlled homomultimerizing module Cry2clust. Upon blue light exposure, Cry2clust forms reversible multimers that constrain the fused CARD9[CARD] domains to a local concentration sufficient to form a polymeric template for BCL10. (**B**) Microscopy images of 293T *BCL10*-KO cells stably expressing CARD9[CARD]-miRFP670nano-Cry2clust and BCL10-mScarlet. Cells were maintained at 37°C with 5% $CO_2$ and imaged every 2 min followed by a short pulse of 488-nm light stimulation. Inset number 1, at 6-min post-stimulation, shows BCL10 colocalizing with a CARD9[CARD] cluster. Inset number 2, at 20 min, shows that the CARD9[CARD] cluster has dissolved, while the BCL10 punctum has instead elongated into a filament. Scale bar: 10 μm. (**C**) Values of the coefficient of variation of mIRFP670nano and mScarlet pixel intensities over time. The data are from 40 cell measurements. Shown are means ± standard deviation (SD). (**D**) Microscopy images of cells expressing CARD9[CARD] seeds, BCL10-mScarlet and EYFP-p65. Images were taken every 2 min following blue light stimulation. For technical reasons, EYFP-p65 was imaged only at the first and last time points. Scale bar: 10 μm. (**E**) Ratios of EYFP fluorescence intensities in the nucleus versus cytosol after blue light stimulation for the indicated durations. Shown are measurements from each cell, along with the means ± standard error of the mean (SEM).

The online version of this article includes the following figure supplement(s) for figure 5:

**Figure supplement 1.** BCL10 nucleation depends on the proper interaction with CARD9 optogenetic clusters.

**Figure supplement 2.** BCL10-mutant E53R fails to nucleate or induce nuclear transcription factor- $\kappa$ B (NF- $\kappa$ B) activity upon CARD9[CARD] optogenetic activation.

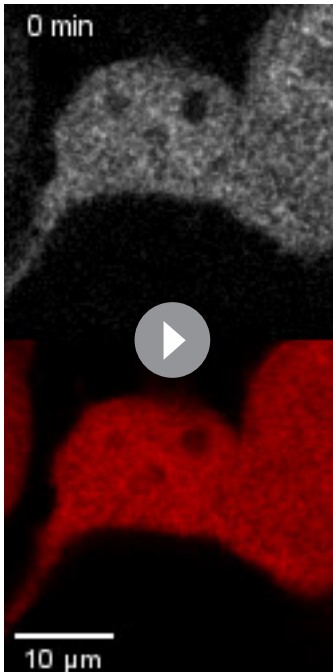

**Video 2.** Optogenetic activation of CARD9<sup>CARD</sup> reveals BCL10 supersaturation. This video shows optogenetic induction of CARD9<sup>CARD</sup>-miRFP670-Cry2 clusters (top panel). After a few frames, CARD9 clusters induce BCL10-mScarlet (bottom panel) nucleation, which goes on to polymerize independently of CARD9 clusters. Thus, revealing a pre-existing state of supersaturation. Scale bar: 10 μm.
https://elifesciences.org/articles/79826/figures#video2

and inset), revealing that BCL10 is driven toward a polymeric steady state even in the absence of CARD9 templates. To test the specificity of the CARD9-BCL10 interaction, we also performed the experiment with the R18W and R70W mutants of CARD9<sup>CARD</sup>-Cry2, whereupon BCL10 did not polymerize despite comparable kinetics of CARD9<sup>CARD</sup>-Cry2 clustering (*Figure 5—figure supplement 1A–D*).

We then asked if the optogenetically nucleated BCL10 polymers sufficed to induce the nuclear translocation of EYFP-p65. In order to avoid repeated stimulation of Cry2 oligomerization with the 514 nm laser, we only imaged EYFP-p65 at the beginning and end (40 min following the blue light pulse) of the experiment. We observed that blue light-induced EYFP-p65 to translocate comparably to stimulation with PMA (*Figure 5D,E*). Additionally, as with the different doses of PMA, the degree of translocation within cells did not depend on the duration of the blue light pulse. This again highlights the all-or-none nature of signaling from BCL10. We found that the polymerization-incompetent mutant of BCL10, E53R, failed to form puncta or activate NF-κB in response to blue light (*Figure 5—figure supplement 2A–C*). Altogether these data indicate that BCL10 is endogenously supersaturated and physiologically restrained from activation by the kinetic barrier associated with polymer nucleation, which collectively allow for switch-like CBM signalosome formation and NF-κB activation.

Throughout our experiments with both THP-1 and HEK293T cells – whether analyzing BCL10 puncta formation, NF-κB transcriptional activity, or p65 translocation – we observed that 25–30% of cells failed to respond to stimulation even at saturating levels of β-glucan or PMA (*Figure 1B,C,E* and *Figure 4E*). The size of this population did not depend on whether BCL10 was endogenously or exogenously expressed, and dropped to 10% in our optogenetic experiments that bypassed upstream factors (*Figure 5—figure supplement 1E*), together suggesting that recalcitrance results at least in part from an increase in the BCL10 nucleation barrier due to cell-to-cell heterogeneity in upstream factors. While the origin of this phenomenon remains to be determined, it demonstrates that cells may regulate their sensitivity to stimulation by raising or lowering the nucleation barrier.

## MALT1 clustering suffices for activation and does not depend on an ordered structure of BCL10 polymers

That BCL10 function derives from a nucleation barrier implies that the structure of the active state has evolved toward increased order relative to the inactive state, irrespective of its interactions with downstream signaling components. If so, the well-ordered nature of BCL10 polymers (as distinct from merely a high density of subunits) should be irrelevant for MALT1 activation. To determine if there is any specific requirement of the BCL10 polymer structure for downstream signaling, we asked if MALT1 can be activated simply by increasing its local proximity in the absence of an ordered scaffold. To do so, we used optogenetic approaches to directly dimerize or multimerize MALT1 lacking its death fold domain. Specifically, we fused MALT1<sup>126-824</sup> to either the blue light-dependent dimerization module, VfAU1-LOV, or the blue light-dependent multimerization module, Cry2clust, followed by the fluorescent protein miRFP670nano (*Figure 6A,B*). When expressed in HEK293T cells containing

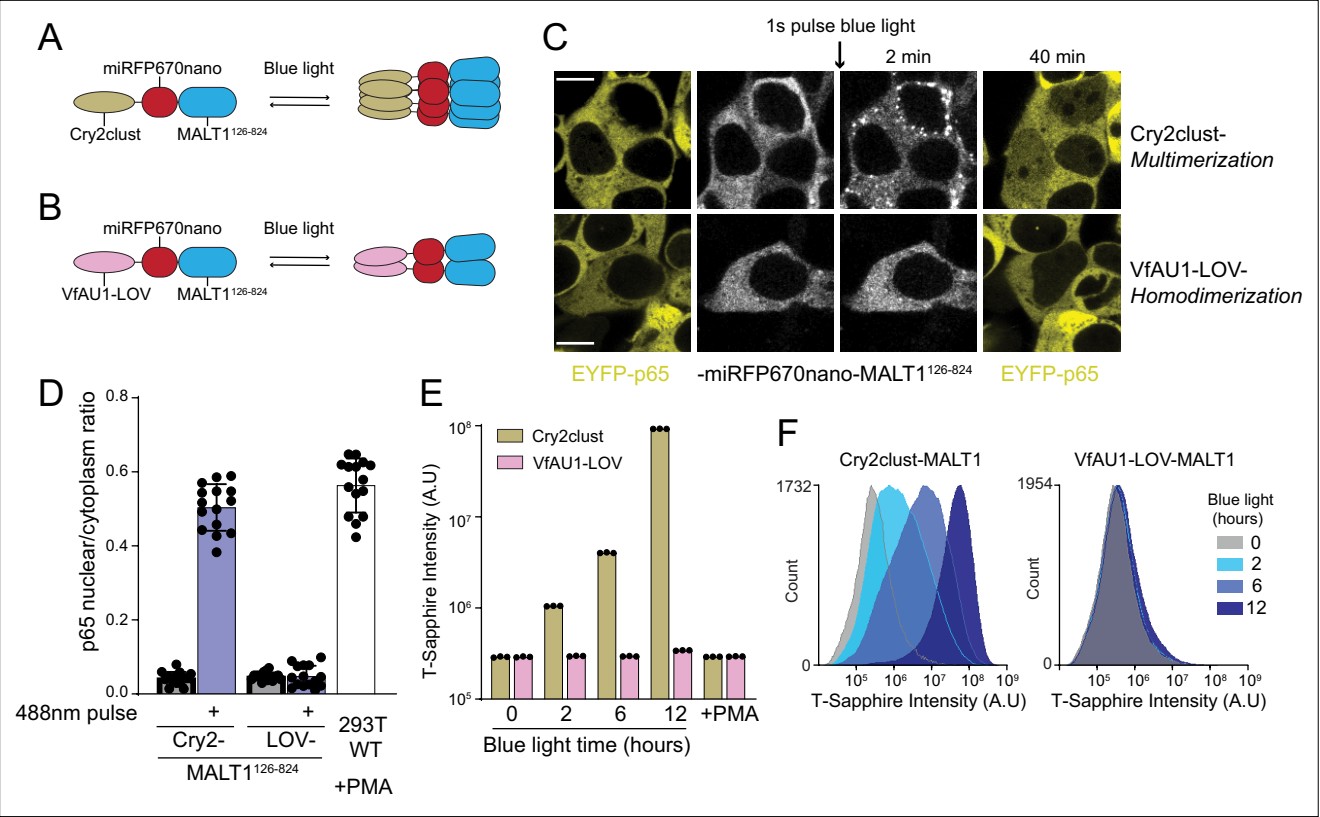

**Figure 6.** MALT1 clustering suffices for activation and does not depend on an ordered structure of BCL10 polymers. (**A**) Schematic of the optogenetic MALT1 activation experiment. A fragment of MALT1 lacking its death domain (MALT1$^{126-824}$) is expressed as a fusion to mIRFP670nano and Cry2clust, allowing for its oligomerization upon blue light exposure. (**B**) Schematic of optogenetic MALT1 dimerization experiment. As for (**A**), but with Cry2clust replaced with the light-controlled dimerizing module, VfAU1-LOV. (**C**) Microscopy images of 293T cells expressing EYFP-p65 transfected with either the construct in (**A**) or (**B**). Images were taken every 2 min following a single pulse of blue light. EYFP-p65 was imaged before blue light stimulation and again 40 min after stimulation, showing that it translocated to the nucleus only in cells in which MALT1 had multimerized. Scale bar: 10 μm. (**D**) Ratios of EYFP fluorescence intensities in the nucleus versus cytosol at the 40 min time point for the experiments in (**C**). As a control, 293T WT expressing EYFP-p65 were treated with PMA 5 ng/ml. Shown are measurements from all cells, along with means ± standard deviation (SD) of three replicates. (**E**) Nuclear transcription factor-κB (NF-κB) activation in 293T *MALT1*-KO NF-κB reporter cells transfected with the constructs in (**A**) and (**B**) and stimulated with continuous blue light for the indicated durations. Cells were treated with PMA at 5 ng/ml. Data represent medians of three independent measurements. (**F**) Flow cytometry histograms of T-Sapphire fluorescence for the experiment in (**E**).

The online version of this article includes the following figure supplement(s) for figure 6:

**Figure supplement 1.** Validation of optogenetic modules using CASP8 triggered cell death.

EYFP-p65 and illuminated with blue light, the dimerizing module failed to induce nuclear transloca-tion, while the multimerizing module did so robustly, and to the same levels as had been achieved for BCL10-mediated activation using HEK293T WT treated with PMA (*Figure 6D*). As a control that the constructs were each functioning as intended, we also performed experiments with the analogous region of CASP8 (180–479) fused to either VfAU1-LOV or Cry2clust. CASP8 has previously been shown to activate upon dimerization (*Demarco et al., 2020*). Indeed, both constructs triggered rampant cell death upon blue light illumination (*Figure 6—figure supplement 1A, B*). Together these results reveal that MALT1 is activated by proximity rather than structural order of the adaptor protein polymers, consistent with BCL10 having a kinetic rather than a specific structural function. The fact that MALT1 activation requires higher-order multimerization rather than just dimerization may relate to its function in concentrating the ubiquitin ligase TRAF6, a critical intermediary in the activation of NF-κB (*Ginster et al., 2017*).

If the binary response of NF-κB to CBM stimuli is indeed conferred by BCL10, then it should switch to a graded response when MALT1 is activated optogenetically. To test this prediction, we expressed the Cry2clust- and VfAU1-LOV MALT1$^{126-824}$ fusions in our NF-κB reporter cell line, and examined the

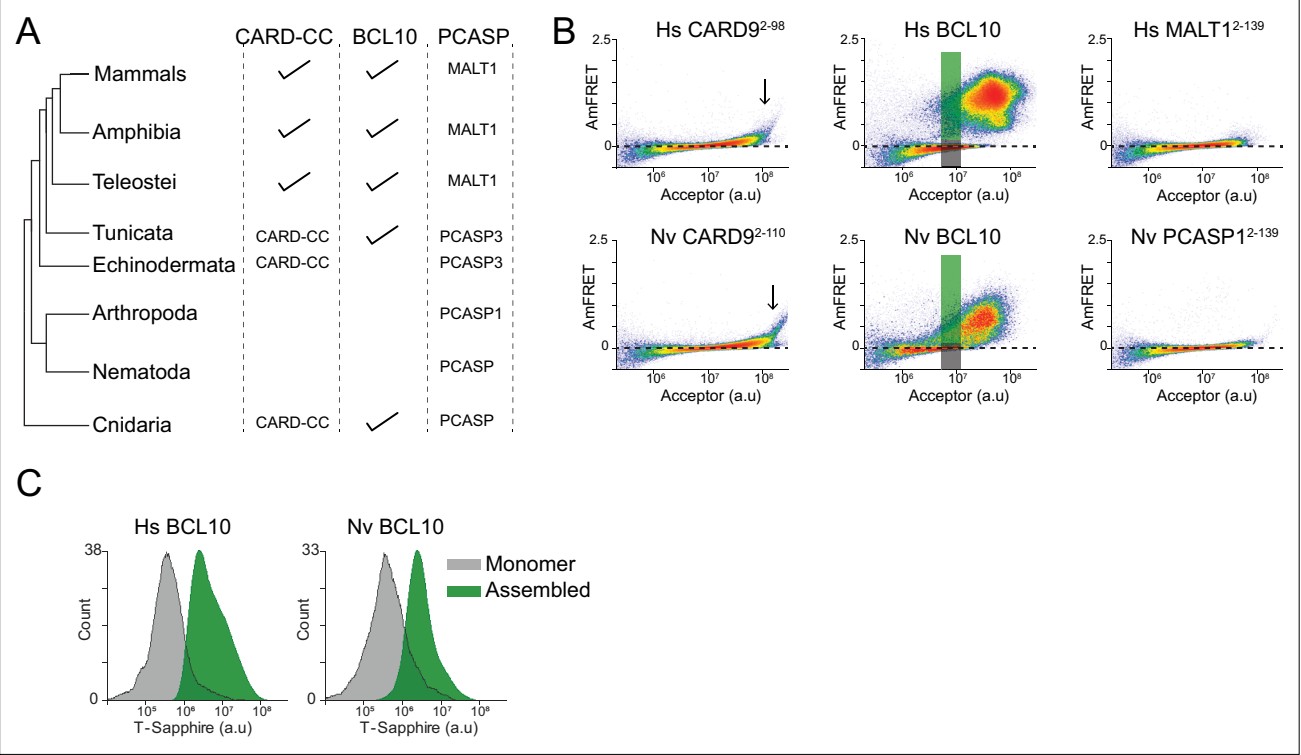

**Figure 7.** Ancient origin of the sequence-encoded nucleation barrier in BCL10. (**A**) Phylogenetic tree of CARD-BCL10-MALT1 (CBM) signalosome components in the major metazoan clades. One or more CARD-CC proteins occur in all clades, as does a MALT1-like paracaspase. BCL10 was present in the common ancestor to chordates and cnidarians. (**B**) DAmFRET plots of 293T cells expressing CARD9[CARD], BCL10, or MALT1[DD] from human (top) or the sea anemone *Nematostella vectensis* (bottom). The gray and green boxes designate the regions gated for cells containing either monomer or polymerized protein, respectively, and analyzed for T-Sapphire expression in (**C**). (**C**) Flow cytometry histograms of T-Sapphire fluorescence in 293T *BCL10*-KO nuclear transcription factor-$\kappa$B (NF-$\kappa$B) reporter cells 48 hr after transfection with either human or *N. vectensis BCL10*-mEos3.2. Gray and green shading represent cells containing either monomeric or polymerized BCL10, respectively, from the corresponding gates illustrated in (**B**). Data are representative of three independent experiments.

The online version of this article includes the following figure supplement(s) for figure 7:

**Figure supplement 1.** The BCL10 CARD domain structure is conserved between human and *N. vectensis* orthologs.

population-level distribution of NF-κB activity in single cells exposed to continuous blue light for different durations. As expected, the intensity of T-Sapphire fluorescence increased in a monotonic fashion with the duration of blue light stimulation (*Figure 6E,F*). When we instead treated these cells with PMA, the NF-κB reporter did not activate (*Figure 6E*). This result indicates that BCL10 polymers formed upon PMA treatment are unable to activate MALT1 when it lacks its DD, ruling out any unanticipated role of the polymers in structuring MALT1[126-824]. These data are consistent with BCL10 conferring the switch-like activation of NF-κB in WT cells upon physiological CBM stimulation.

## Ancient origin of the sequence-encoded nucleation barrier in BCL10

The CBM signalosome originated in the ancient ancestor of cnidarians and bilaterians (*Staal et al., 2018*; *Figure 7A*). To determine if binarization is a conserved ancestral function of this signaling pathway, we investigated the structure and assembly properties of BCL10 from the model cnidarian, *Nematostella vectensis* (Nv). Using AlphaFold2 (*Jumper et al., 2021*), we found that the overall structure of the CARD, predicted with high confidence, is highly similar to that of human BCL10, and retains key residues of the polymer interface, such as E53 (*Figure 7—figure supplement 1A–C*).

We next expressed each of the Nv CBM components as mEos3.2 fusions in human cells. We found that the DAmFRET profiles for all three proteins are strikingly similar to those of their human counterparts. Specifically, the CARD of CARD-CC, the CARD9 homolog, polymerized with a high saturating concentration and negligible nucleation barrier (*Figure 7B*). Nv BCL10 exhibited a discontinuous,

nucleation-limited transition between monomer and polymer, and the death fold from Nv PCASP1, the MALT1 homolog, remained entirely monomeric (*Figure 7B*).

Finally, we asked if NvBCL10 can functionally replace human BCL10 with respect to binary activation of NF-κB. We expressed the protein in HEK293T *BCL10*-KO cells containing the transcriptional reporter (T-Sapphire) of NF-κB activation. By measuring T-Sapphire fluorescence within a DAmFRET experiment, we found that indeed, the population of cells with NvBCL10 polymers also contained activated NF-κB, while cells expressing the same concentration of NvBCL10 in its monomeric form contained inactive NF-κB (*Figure 7C*). Together, these results suggest that the function of human BCL10 as a kinetic determinant of cellular decisions has been retained for over 600 million years of animal evolution.

## Discussion

In this study, we investigated the influence of CBM signalosome assembly on the binary kinetics of transcription factor NF-κB activation, a crucial effector of immune signaling. Using biophysical tools and a single-cell reporter of NF-κB activity, we revealed that the all-or-none response of NF-κB has a physical origin in the CBM signalosome, and that it specifically results from an ancient sequence-encoded nucleation barrier in the CARD of the adaptor protein, BCL10. Our discovery that human CBM mutations that are responsible for pathogen susceptibility and cancer, respectively, increase or decrease the barrier suggests that it has been evolutionary tuned for optimum immune system function.

This barrier allows BCL10 molecules to exist in a physiologically supersaturated state. In this state, the protein molecules are soluble and inactive, yet poised for mass activation when even a tiny fraction of them are templated to the polymer conformation by stimulated oligomers of a CARD-CC protein. We characterized in depth one of those proteins – CARD9 – and showed that its long coiled-coil region drives it into large multimers that facilitate nucleation by lowering the entropic cost to assembling multiple CARD subunits into a nucleating configuration. We reiterate that the role of upstream CARD-CC proteins is primarily kinetic in nature and culminates with the initial BCL10-nucleating event, as evidenced by our finding that BCL10 continues to polymerize even after the experimental dissolution of CARD nuclei. Nevertheless, our data do not imply that the frequency or duration of CBM formation cannot be regulated. Indeed, our observations of cells recalcitrant to stimulation show that the nucleation barrier can differ between cells. We suspect that some of the known regulatory modifications to CBM and upstream proteins function to tune the nucleation barrier. Others represent negative feedback that eventually allows signaling to terminate (*Ruland and Hartjes, 2019*; *Thome and Weil, 2007*).

CBM is one of multiple immune signalosomes featuring ordered polymers that, collectively, have been conserved since the dawn of multicellular immune systems (*Dyrka et al., 2020*; *Essuman et al., 2018*; *Saupe, 2020*; *Wu and Fuxreiter, 2016*). The prevailing notions for the functions of these polymers, which can be broadly categorized as ultrasensitivity (threshold responsiveness) and proximity-dependent effector activation (*Matyszewski et al., 2018*; *Nanson et al., 2019*; *Vajjhala et al., 2017*; *Wu and Fuxreiter, 2016*) do not encompass bistability. The former activities emerge from the equilibrium properties of phase separation, whereas the latter necessarily involves a kinetic property. We previously showed that nucleation barriers large enough to partition identically stimulated cells into discrete states only occur when phase separation is coupled to a conformational transition (*Khan et al., 2018*; *Posey et al., 2021*; *Rodríguez Gama et al., 2021*). The preponderance of ordered polymers in immunity signalosomes is therefore consistent with their functioning to confer bistability rather than threshold sensitivity, consistent with the need for immune systems to respond decisively to vastly substoichiometric stimulation. This is further evidenced by the fact that (1) proximity-dependent effector activation does not require any specific polymer structure, as shown presently for MALT1 and previously for caspases-1 and -8 (*Boucher et al., 2018*; *Shen et al., 2018*; *Würstle et al., 2010*); and (2) polymers have finite structural nuclei that limit cooperativity (*Khan et al., 2018*; *Vekilov, 2012*) whereas phase separation allows for theoretically infinite cooperativity (*O'Flynn and Mittag, 2021*; *Rodríguez Gama et al., 2021*). Indeed, signaling networks commonly feature liquid–liquid phase separation without any apparent transition in structural order (*Dignon et al., 2020*; *Gibson et al., 2019*; *Riback et al., 2017*; *Yoo et al., 2019*).

In our view, signalosome polymers are evolutionary spandrels (*Gould and Lewontin, 1979*; *Manhart and Morozov, 2015*) – byproducts rather than primary targets of natural selection. They are not well described by the classical protein structure–function paradigm, that is, that a protein's function emerges directly from a particular three-dimensional structure (*Redfern et al., 2008*; *Branden and Tooze, 2012*; *Papoian, 2008*). This is because BCL10 function emerges from a kinetic property of its collective that can be conferred by any sufficiently restrictive entropic bottleneck. That activated BCL10 is an ordered polymer appears to be a consequence of selection for that bottleneck more than for some activity of the polymer itself. This is plausibly why signalosomes have evolved from at least three structurally distinct polymer scaffolds, including bona fide amyloids that in most other contexts cause protein dysfunction (*Rodríguez Gama et al., 2021*). Any sequence variant that flattens the nucleation barrier for example by preordering the monomers or disordering the polymers, is likely to reduce the sensitivity and executive functionality of the signalosome. As opposed to liquid-like non-membrane-bound organelles, extant signalosomes have therefore evolved such a high level of structural order that they do not form spontaneously (or only rarely so) despite a thermodynamic driving force to do so. In short, our results imply that a protein's structure can evolve via selection for kinetic rather than equilibrium properties of its collective.

Nevertheless, nucleation is inherently probabilistic and, with enough time, may occur even in the absence of stimulation. In embracing supersaturation to power immune signaling, cells may have also predetermined their fates. That BCL10 and potentially other pro-inflammatory signalosome proteins are constitutively supersaturated implies that cells are thermodynamically predisposed to inflammation, and therefore hints at an ultimate physical basis for the centrality of inflammation in progressive and age-associated diseases (*Furman et al., 2019*; *López-Otín et al., 2013*; *Taniguchi and Karin, 2018*). This fact should guide ongoing efforts to control inflammation for the betterment of human health.

# Materials and methods

## Key resources table

| Reagent type (species) or resource | Designation | Source or reference | Identifiers | Additional information |
|---|---|---|---|---|
| Cell line (*Homo sapiens*) | HEK293T | American Type Culture Collection | # CRL-3216 | |
| Cell line (*Homo sapiens*) | THP-1 | American Type Culture Collection | # TIB-202 | |
| Strain, strain background (*Saccharomyces cerevisiae*) | rhy1713 | *Khan et al., 2018* | | Parent strain for DAmFRET experiments |
| Strain, strain background (*Saccharomyces cerevisiae*) | rhy1734 | *Khan et al., 2018* | | Parent strain for DAmFRET experiments |
| Transfected construct (*Homo sapiens*) | m1.0151 | This paper | | Plasmid for expressing Hs CARD9-(G4S)-mCardinal from CMV promoter |
| Transfected construct (*Homo sapiens*) | m1.0012 | This paper | | Plasmid for expressing Hs BCL10-(EA3R)4-mEos3.2 from CMV promoter |
| Antibody | anti-BCL10 (Rabbit polyclonal) | Cell Signaling Technology | C78F1 | Simple Western (1:10) |
| Antibody | anti-CARD9 (Mouse monoclonal) | Santa Cruz Biotechnology | B-12: sc-46677 | Simple Western (1:150) |
| Recombinant DNA reagent | pCW57.1 | Addgene | #41393 | |
| Recombinant DNA reagent | pLV-EF1a-IRES-Hygro | Addgene | #85134 | |
| Chemical compound, drug | PMA | BioVision | 1544–5 | |
| Software, algorithm | GraphPad Prism | Version 9 for Windows | | |

## Antibodies and reagents

Antibodies used in this study include: BCL10 Rabbit polyclonal Ab (Cell Signaling Technology, C78F1), CARD9 mouse monoclonal Ab (Santa Cruz Biotechnology, A-8: sc-374569), and MALT1 mouse monoclonal Ab (Santa Cruz Biotechnology, B-12: sc-46677). β-Actin mouse monoclonal Ab (Santa Cruz Biotechnology, C4: sc-47778). mEos3.1 Rabbit polyclonal Ab (Halfmann laboratory, #4530). Alexa-488

Goat anti-Rabbit IgG (Thermo Fisher Scientific, A32731). Anti-mouse IgG Ab HRP (Cell Signaling Technology, 7076), mouse anti-rabbit IgG-HRP (Santa Cruz Biotechnology, sc-2357). Other reagents include: FuGENE HD (promega, E2311), β-glucan peptide (Invivogen, tlrl-bn, ant-zn-05), Hygromycin B (Invivogen, ant-hg-1), Penicillin–Streptomycin (Thermo Fisher, 1514014 gp), LPS (Sigma-Aldrich, L2880), PMA (BioVision, 1544–5), Halt protease inhibitor (Thermo Fisher, 78439), Puromycin (Invivogen, ant-pr-1), and Zeocin (Invivogen, ant-zn-05).

## Plasmid construction

A list of plasmids used in this study can be found in *Supplementary file 1*. Yeast expression plasmids were made as previously described in *Khan et al., 2018*. Briefly, we used a Golden Gate cloning-compatible vector V08 which contains inverted BsaI sites followed by 4x(EAAAR) (a rigid linker to minimize potential impacts of mEos3.1 on the solution phase ensemble of the protein of interest). V08 vector drives expression of proteins from a GAL promoter and contains the auxotrophic marker URA3. The m1 mammalian expression vector was constructed in two steps from mEos3.2-N1 (Addgene #54525). First an existing BsaI site in mEos3.2-N1 was removed by introducing a point mutation at nucleotide position (3719) G to A. Second, the existing multiple cloning site in mEos3.2-N1 was replaced via Gibson assembly with a fragment containing inverted BsaI sites followed by 4x(EAAAR) to create the Golden Gate compatible vector, m1. In this vector protein expression is controlled by a CMV promoter. m1 was then used to create the m1_C1 vector in two steps. First, inverted BsaI sites upstream of mEos3.1 were removed. Second, using Gibson assembly, a fragment containing 4x(EAAAR) followed by inverted BsaI was inserted in frame after mEos3.2. The mCardinal vector was made via Gibson by replacing the mEos3.2 in m1 with a synthetic fragment encoding mCardinal. Vector m16 was made via Gibson by replacing the mEos3.2 in m1 with a synthetic fragment encoding mScarlet-I. The m25 backbone was made by replacing the mCherry sequence in mCherry-CRY2clust (Addgene #105624) with an insert containing inverted BsaI sited followed by 4x(EAAAR) and miRFP670nano.

Lentivirus vectors were made as follows. The lentivirus vector containing the NF-κB reporter, phage_NFkB-TSapp, was made by replacing the luciferase sequences with the fluorescent protein T-Sapphire in pHAGE NFKB-TA-LUC-UBC-dTomato-W (Addgene #49335). Additionally, we replaced the marker tdTomato in Addgene #49,335 with the resistance cassette for HygromycinB. The lentivirus vector for expression of EYFP-p65, pLV_EYFP-p65 was made by replacing the Hygromycin B resistance cassette in pLV-EF1a-IRES-Hygro (Addgene #85134) with a Zeocin resistance cassette and inserting the p65 coding sequence before the IRES site. To create lentiviral vectors for the expression of CARD9 fused with miRFP670 and Cry2, we inserted via Gibson the respective inserts from m25 into pLV-EF1a-IRES-Hygro. Finally, for the doxycycline controlled lentiviral vectors, we cloned the respective coding sequences from *BCL10* in m1 or m16 vectors into pCW57.1 (Addgene #41393).

Inserts were ordered as GeneArt Strings (Thermo Fisher) flanked by Type IIs restriction sites for ligation between BsaI sites in V08, V12, m1, and other vectors derived from m1. All other inserts were cloned into respective vectors via Gibson assembly between the promoter and respective fluorescent or tag marker. All plasmids were verified by Sanger sequencing.

## Yeast strain construction

For DAmFRET experiments we employed strain rhy1713, which was made as previously described in *Khan et al., 2018*. To create artificial intracellular seeds of CARD9 variants, CARD10, CARD11, and CARD14, sequences were fused to a constitutive condensate-forming protein, μNS (471–721), hereafter 'μNS'. Yeast strain rhy2153 was created by replacing the HO locus in rhy1734 with a cassette consisting of: natMX followed by the tetO7 promoter followed by counterselectable URA3 ORFs derived from C. albicans and K. lactis, followed by μNS-mCardinal. To create strains expressing the fusion CARD9-mCardinal-μNS and others, AseI digests of yeast expression plasmids containing desired proteins were transformed into rhy2153 to replace the counterselectable URA3 ORFs with the gene of interest. The resulting strains express the proteins of interest fused to μNS-mCardinal, under the control of a doxycycline-repressible promoter. Transformants were selected for 5-FOA resistance and validated by PCR. See *Supplementary file 2* for a list of all strains used in this study.

For measuring nucleating interactions diploid strains were maintained on doxycycline (40 mg/ml) until initial culture for DAmFRET assay.

## Cell culture

HEK293T (CRL-3216) cells and THP-1 (TIB-202) cells were purchased from ATCC. HEK293T cells were grown in Dulbecco's modified Eagle's medium (DMEM) with L-glutamine, 10% fetal bovine serum (FBS), and PenStrep 100 U/ml. THP-1 cells were grown in Roswell Park Memorial Institute (RPMI) medium 1,640 with L-glutamine and 10% FBS. All cells were grown at 37°C in a 5% $CO_2$ atmosphere incubator. Cell lines were regularly tested for mycoplasma using the Universal mycoplasma detection kit (ATCC, #30-1012K).

## Transient transfections

HEK293T cells were plated in 6-well culture plates at $8.0 \times 10^5$ cells/well in DMEM. The next day, 2.0 µg DNA of expression plasmids were mixed 150 µl Opti-MEM and transfected using FuGENE HD (Promega) at a DNA to FuGENE HD ratio of (1:3). Cells were treated with either PMA or Dimerizer ligand at described concentrations 24 hr after transfection. Cells were harvested for flow cytometry analysis after a total of 48 hr. For microscopy experiments, cells were plated directly into 35 mm glass-bottom dishes (iBidi, µ-Dish 35 mm, high Glass Bottom) at a concentration of $0.8 \times 10^5$ cells/dish and transfected as previously mentioned.

## Generation of stable cell lines

For generation of stable cell lines, constructs were packaged into lentivirus in a 10 cm plate 60% confluent of HEK293T cells using the TransIT-LT1 (Mirus Bio, MIR2300) transfection reagent and 7 µg of the vector, 7 µg psPAX2, and 1 µg pVSV-G. After 48 hr, media was collected and centrifuged at $3000 \times g$ to remove cell debris. At this point, media containing lentivirus was used or stored at −80°C. For HEK293T adherent cell transduction, cells were plated in 6-well culture plates at 50% density. Next day, media was replaced with media containing lentivirus at a multiplicity of infection of 1, supplemented with 5 µg/ml Polybrene (Sigma-Aldrich, TR-1003-G). For THP-1 suspension cell transduction, cells were spinfected with virus containing media for 1 hr at $1000 \times g$ at room temperature supplemented with 5 µg/ml Polybrene. For transduction of phageNF-κB-TSapp, THP-1 and HEK293T cells were selected with Hygromycin B (350 mg/ml) for 14 days and used for further analysis or complementary transductions. For transduction of pCW57.1_BCL10-mScarlet, pwtBCL10-mScarlet, HEK293T cells were selected with Puromycin (1 µg/mL) for 7 days. After this time, cells were sorted for positive expression of mScarlet and expanded in continuing selection with puromycin. For transduction of expression clones tagged with mEos3.1, THP-1 cells were selected with Puromycin (1 µg/ml) for 7 days. Cells were sorted for positive expression of mEos3.1 and expanded for further experiments with continued selection in puromycin. For transduction of pLV_EYFP-p65, HEK293T *BCL10*-KO cells reconstituted with pBCL10-mScarlet, were transduced and selected with Zeocin (300 µg/ml). Positive cells expressing EYFP and mScarlet were sorted and expanded in selection media containing drugs. Plasmid pCARD9-CARD_miRFP-Cry2 was transduced into HEK293T cells. Cells were selected with Hygromycin B (350 µg/ml) for 7 days and then sorted for positive expression of miRFP670nano.

## CRISPR-Cas9-mediated knockout

To generate HEK293T *BCL10*-KO and MALT KO cell line, cells were nucleofected (Neon Transfection system, Thremo) with a ribonucleoprotein (RNP) mix of sgRNA for *BCL10* exon1 or MALT1 exon1 and Cas9 protein. After this, cells were pooled and submitted for targeted deep sequencing to find the insertion/deletion (InDel) ratio. Subsequently, cells were prepared for single-cell sorting into four 96-well plates and expanded to obtain a duplicate for each well. At this stage cells were lysed and submitted for deep sequencing to find the genome edited cells. Selected wells were further expanded to validate via protein immuno detection. To generate THP-1 *BCL10*-KO cells, sgRNA targeting *BCL10* exon1 was cloned into the lentiCRISPR v2-Blast (Addgene #83480). This vector was packaged into lentivirus as described in generation of stable cell lines section. THP-1 cells were transduced via spinfection with lentivirus and supplemented with polybrene. 24 hr after spinfection, media was replaced. 48 hr after spinfection, cells were selected with blasticidin (1 µg/ml). After 10 days of blasticidin selection, single-cell clonal expansion was done by serial dilution of resistant cells to achieve complete knockouts. Selected wells were analyzed by immuno detection to confirm the absence of BCL10 protein.

### Yeast preparation for DAmFRET

We performed DAmFRET analysis as previously described in *Khan et al., 2018*. Expression plasmids were transformed into specified strains following a standard lithium acetate protocol (see *Supplementary file 1* for list of expression plasmids). Yeast transformants were selected in SD-URA plates. Briefly, single transformant colonies were inoculated in 200 µl of SD-URA in a microplate well and incubated in a Heidolph Titramax platform shaker at 30°C, 1350 RPM overnight without presence of Dox. Cells were washed with sterile water, resuspended in galactose-containing media, and allowed to continue incubating for approximately 16 hr. Microplates were then illuminated for 25 min with 320–500 nm violet light to photoconvert an empirically optimized fraction of mEos3.1 molecules from a green (516 nm) form to a red form (581 nm). At this point, cells were either used to collect microscopy data or continue the DAmFRET protocol.

### Mammalian cell preparation for DAmFRET

We performed DAmFRET in HEK293T cells as previously described (*Venkatesan et al., 2019*). Briefly, cells were transfected as described in the methods section on transient transfection. 48 hr after transfection, cells were trypsinized and washed in phosphate-buffered saline (PBS) supplemented with 10 mM ethylenediamine tetraacetic acid (EDTA). Then, samples were fixed in 4% paraformaldehyde (PFA) for 5 min with constant shaking. Cells were washed two additional times with PBS + EDTA and transferred to a 96-well plate. The sample-containing plate was photoconverted as previously described for yeast samples. After photoconversion, samples were analyzed in a ZE5 cell analyzer cytometer.

### DAmFRET cytometric data collection

DAmFRET data were collected on a ZE5 cell analyzer cytometer or an ImageStreamx MkII imaging cytometer (Amnis) at ×60 magnification as previously described (*Khan et al., 2018*). Autofluorescence was detected with 405 nm excitation and 460/22 nm emission; Side Scatter (SSC) and Forward Scatter (FSC) were detected with 488 nm excitation and 488/10 nm emission. Donor and FRET fluorescence were detected with 488 nm excitation and 525/35 nm or 593/52 nm emission, respectively. Acceptor fluorescence was excited with 561 nm excitation and 589/15 nm emission. Approximately 500,000 events were collected per sample. Data compensation was done in the built-in tool for compensation (Everest software V1.1) on single-color controls: non-photoconverted mEos3.1 and dsRed2 (as a proxy for the red form of mEos3.1). For nucleating interactions, all regular channels for DAmFRET were collected with the addition of mCardinal intensity by 561 nm excitation and 670/30 nm emission.

DAmFRET coupled with the NF-κB reporter data was collected as previously mentioned with the following modifications. HEK293T cells were treated as previously described in the DAmFRET protocol. In addition to the regular channel for DAmFRET data acquisition, T-Sapphire expression was detected using 405 nm excitation and 525/50 nm emission.

### DAmFRET analysis

Data were processed on FCS Express Plus 6.04.0015 software (De Novo). Events were gated for single unbudded cells by FSC versus SSC, followed by gating of live cells with low autofluorescence and donor positive. Live gate was then selected for double positives (donor and acceptor). Plots represent the distribution of AmFRET (FRET intensity/acceptor intensity) versus Acceptor intensity (protein expression). For DAmFRET data in mammalian cells, cells were first gated for single cells followed by gating double positive expressing cells donor, acceptor. In the cases for DAmFRET coupled to NF-κB reporter, an additional gating was performed on the DAmFRET plot to quantify the intensity of T-Sapphire.

The data were fit to a Weibull function as previously described (*Khan et al., 2018*). Briefly, cells exhibiting assemblies were identified by their exclusion from a negative DAmFRET gate created from the monomeric mEos3.1 distribution. DAmFRET plots were divided into 64 logarithmically spaced bins and, within each, the fraction of cells with AmFRET higher than the monomer control was calculated. We applied this approach to all DAmFRET plots. This analysis provides a gross percentage of the fraction of cells containing assembled protein. The fit of these data to the Weibull function produced the $EC_{50}$ and $\delta$ statistics.

## Fluorescence microscopy

Imaging of yeast and mammalian cells was done in an LSM 780 microscope with a ×63 Plan-Apochromat (NA = 1.40) objective. T-Sapphire excitation was done with a 405 nm laser. mEos3.1 and mScarlet-I excitation were made with a 488 and 561 nm laser, respectively. To quantify the percentage of cells containing polymers, images were subjected to an in-house Fiji (https://imagej.net/software/fiji/) adapted implementation of Cellpose (https://www.nature.com/articles/s41592-020-01018-x, https://www.cellpose.org/) analysis for cellular segmentation for both whole cell contour and nucleus. The Cellpose generated regions of interest (ROIs) were used to measure specified imaging channels. For each experiment, more than 200 cells were quantified in an unbiased way. Cells were defined as positive for containing polymers if the coefficient of variation (CV, standard deviation divided by the mean intensity) for the fluorescent channel was above 0.8 as defined by the distribution of values.

For time-lapse imaging, fluorescence images were taken using a spinning-disk confocal microscope (Nikon, CSU-W1) with a ×60 Plan Apochromat objective (NA = 1.40) and a Flash 4 sCMOS camera (Hamamatsu). Samples were maintained at 37°C and 5% $CO_2$ with a stage top incubator (Okolab). Prior to stimulation, cells were treated with Hoechst 33,342 (0.5 µg/ml) for 30 min. 405 nm laser at 20% was used to collect emission of Hoechst for 50 ms per frame. 514 and 561 nm laser were used at 25% to collect EYFP and m-Scarlet emission for 300 ms, respectively. Images were captured every 10 min for p65 nuclear translocation experiments. ROIs were generated by Cellpose segmentation algorithm around each cell contour and nucleus on the EYFP and Hoechst image, respectively. These ROIs were then used to measure the area, mean, standard deviation, and integrated density of each cell on the EYFP and mScarlet fluorescence channels. For each cell, we computed the nuclear to cytoplasmic ratio of integrated density for the EYFP channel. The same ROIs were used to compute the CV of the mScarlet-I fluorescence.

## Optogenetic activation

Cry2clust optogenetic experiments were conducted on an LSM 780 microscope with a ×63 Plan-Apochromat (NA = 1.40) objective in the integration mode. Samples were maintained at 37°C and 5% $CO_2$ with a microscope incubator. To stimulate Cry2clust we used the 488 nm laser at a power setting of 50% for a pulse of 1 s, which is the amount of time it took to scan the user generated region of interest, unless indicated otherwise. 561 and 633 nm lasers were used for imaging mScarlet-I and miRFP670nano, respectively. To track BCL10 nucleation dependent on CARD9-Cry2 we imaged cells before and after stimulation every 2 min. For the experiments to quantify the nuclear translocation of p65 to the nucleus, EYFP was imaged using the 514 nm laser. Imaging EYFP with the 514 laser resulted in undesired Cry2clust restimulation. To avoid additional Cry2clust stimulation in our experiments, EYFP was only imaged at the beginning of the experiment and in the last frame. We computed the nuclear to cytoplasmic ratio of EYFP as previously described in the section above. Similarly, we calculated the CV for mScarlet-I intensity as mentioned above.

## Immunostaining

To perform immunostaining of NF-κB subunit, p65, first THP-1 BCL10-KO cells reconstituted with either BCL10-mScarlet or BCL10(E53R)-mScarlet were seeded on glass coverslips (Neuvitro, GG-18-PLL) placed on 12-well plates at a density of $3.0×10^5$ cells/well. PMA was added at a concentration of 10 ng/ml for 24 hr. This allowed cells to attach on the coverslips. After withdrawal of PMA, cells rested for 24 hr and then BCL10-mScarlet protein expression was induced with doxycycline (0.8 µg/ml) for 16 hr. Dox was removed from the wells and replaced with fresh media. Samples were then treated with β-glucan (10 µg/ml) for 12 hr. Prior primary antibody incubation, cells attached to the coverslips were washed with PBS and then fixed with PFA 4% for 15 min. PFA was then washed away with PBS rinse and added PBS + Triton 0.05% to permeabilize cells. Cells were then washed three times with PBS and blocked with bovine serum albumin 1% for 30 min at room temperature. Primary antibody incubation ocurred overnight with p65 antibody (Proteintech, 10745-1-AP) at a dilution of 1:400. Primary antibody was washed three times with PBS. After this, cells were incubated with anti-rabbit Alexa Fluor 647 secondary antibody (Thermo Fisher, A-31573) at a 1:400 dilution for 4 hr at RT. Following secondary antibody wash steps, cells were stained with DAPI (0.1 µg/ml) for 10 min. Subsequently, cells were mounted on glass slides using ProLong Gold (Thermo Fisher, P36965). Images were collected in LSM 780 microscope with a ×63 Plan-Apochromat (NA = 1.40) objective.

## Protein immunodetection

The Wes platform (ProteinSimple) was used to perform capillary based protein immunodetection. We followed the recommended protocol from the manufacturer. Briefly, cell lysates were diluted with 0.1× sample buffer. Subsequently, four parts of diluted sample were combined with one part 5× Fluorescent Master Mix (containing 5× sample buffer, 5× fluorescent standard, and 200 mM dithiothreitol (DTT)) then boiled at 95°C for 5 min. Our sample final protein concentration for each capillary was 1 µg/ml. After the denaturing step, an assay plate was filled with samples, blocking reagent, primary antibodies (1:50 dilution for mEos3.1, MALT1 and actin, 1:10 dilution for BCL10, and 1:150 dilution for CARD9), HRP-conjugated secondary antibodies and chemiluminescent substrate. A biotinylated ladder provided molecular weight standards for each assay. Once the assay plate was set up, electrophoretic protein separation and immunodetection were carried out in the fully automated capillary system. Data were processed using the open-source software Compass (https://www.proteinsimple.com/compass/downloads/) to extract the intensities for the peaks corresponding to the expected molecular weight of proteins of interest.

## Quantification and statistical analysis

For two sample comparisons, two-sided Student's *t*-tests were used for significance testing unless stated otherwise. The graphs represent the means ±standard deviation of independent biological experiments unless stated otherwise. Statistical analysis was performed using GraphPad Prism 9 and R packages.

## Acknowledgements

We thank Xiaoqing Song for assistance with immunostaining experiments, and members of the Halfmann lab for critical reading of the manuscript. We are also grateful to the very constructive anonymous reviewers of *Review Commons*. This work was performed to fulfill, in part, requirements for ARG's thesis research in the Graduate School of the Stowers Institute for Medical Research. This work was supported by the National Institute of General Medical Sciences (Award Number R01GM130927, to RH) and the National Institute on Aging (Award Number F99AG068511, to ARG) of the National Institutes of Health, the American Cancer Society (RSG-19-217-01-CCG to RH), and the Stowers Institute for Medical Research. The funders had no role in study design, data collection and analysis, or manuscript preparation. The content is solely the responsibility of the authors and does not necessarily represent the official views of the funders. Original data underlying this manuscript can be accessed from the Stowers Original Data Repository at http://www.stowers.org/research/publications/libpb-1675.

## Additional information

### Funding

| Funder | Grant reference number | Author |
| --- | --- | --- |
| National Institute of General Medical Sciences | R01GM130927 | Randal Halfmann |
| National Institute on Aging | F99AG068511 | Alejandro Rodriguez Gama |
| American Cancer Society | RSG-19-217-01-CCG | Randal Halfmann |
| Stowers Institute for Medical Research | | Jay R Unruh Randal Halfmann |

The funders had no role in study design, data collection, and interpretation, or the decision to submit the work for publication.

### Author contributions

Alejandro Rodriguez Gama, Conceptualization, Formal analysis, Funding acquisition, Investigation, Methodology, Visualization, Writing – original draft, Writing – review and editing; Tayla Miller, Jeffrey

J Lange, Formal analysis, Investigation, Writing – review and editing; Jay R Unruh, Formal analysis, Writing – review and editing; Randal Halfmann, Conceptualization, Funding acquisition, Methodology, Supervision, Writing – original draft, Writing – review and editing

**Author ORCIDs**
Alejandro Rodriguez Gama (ID) http://orcid.org/0000-0003-3257-5549
Jay R Unruh (ID) http://orcid.org/0000-0003-3077-4990
Randal Halfmann (ID) http://orcid.org/0000-0002-6592-1471

**Decision letter and Author response**
Decision letter https://doi.org/10.7554/eLife.79826.sa1
Author response https://doi.org/10.7554/eLife.79826.sa2

## Additional files

**Supplementary files**
• Transparent reporting form

• Supplementary file 1. List of plasmids and the amino acid sequences of the relevant ORFs used in this study.

• Supplementary file 2. List of *S. cerevisiae* strains used in this study.

**Data availability**
Original data underlying this manuscript can be accessed from the Stowers Original Data Repository at http://www.stowers.org/research/publications/libpb-1675.

The following dataset was generated:

| Author(s) | Year | Dataset title | Dataset URL | Database and Identifier |
|-----------|------|---------------|-------------|-------------------------|
| Rodriguez Gama A, Miller T, Lange JJ, Unruh JR, Halfmann R | 2022 | A nucleation barrier spring-loads the CBM signalosome for binary activation | https://www.stowers.org/research/publications/libpb-1675 | Stowers Original Data Repository, libpb-1675 |

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
