## [Editor Report]

Signalosomes are multi-protein complexes that transduce signals by sequentially assembling filamentous oligomers. Here, Rodriguez-Gama and colleague present their exciting finding in which the BCL10 adaptor in the CBM signalosome acts as an analogue-to-digital converter, resulting in binary activation of immune cells.

---

## [Decision Letter]

[Editors' note: this paper was reviewed by Review Commons.]

---

## [Author Response]

Reviewer #1 (Evidence, reproducibility and clarity (Required)):In this manuscript, Gama et al. use a biophysical assay DAmFRET, structural analysis, and optogenetic tools to uncover the nucleation mechanism of CBM signalosome. They performed experiments first in yeast cells that lack death folds or related signaling networks, then confirmed their discoveries in human cells. The results presented here are clear and convincing. The paper is very well presented and clearly written.They found it is the CARD domain of BCL10 that acts as a molecular switch that drives all-or-none activation of NF-κB. Monomeric BCL10 possesses an unfavorable conformation and serves as a nucleation barrier, keeping BCL10 in a supersaturated inactive state that allows for binary activation upon stimulation.They also characterized CARD9 CARD domain and a coiled-coil region. They reasoned that CARD9CARD functions as a polymer seed to nucleate BCL10, and that the coiled-coil region has multimerization ability to facilitate nucleation. Furthermore, they characterized that MALT1 activation doesn't depend on BCL10 polymers but its own proximity. And MALT1 induces graded NF-κB activation, thus further demonstrating the binary activation is conferred by BCL10.Major comments:1. Figure S1D and E, the authors used TNF-a to activate NF-κB independent of CBM signalosome and found the activation in each cell increased with dose. In contrast, CBM activation led to bimodal cell activation. The authors claim that this is evidence that positive feedback upstream of NF-κB. We do not believe this claim can be made from this comparative experiment alone. We agree that positive feedback is important for activating an NF-κB response, but the comparison between CBM and TNFa is inaccurate and glosses over published data. Specifically, there is published data that TNF-a does activate a 'switch-like' or digital response, as defined by the translocation of p65 (see (Tay et al. 2010) among other studies that have examined p65 translocation at the single-cell level). The difference in T-sapphire expression between CBM and TNF activation is most likely due to TNFa induced oscillations of p65 translocation (although this is speculation on our part). Therefore we suggest to the authors that the TNF-a data (Figure S1D and E) should be omitted, as the claim of switch or not-switch as pertains to TNF signaling is more complex and nuanced than presented here. We believe omitting this data will strengthen the manuscript and avoid confusion in the field. The bimodal expression of the T-sapphire NF-κB reporter driven by the CBM signalosome activation is sufficient to claim an all-or-none response.

We thank the reviewer for this suggestion. We acknowledge that the activation of NF-κB by TNF-ɑ is more complex than we had presented, and agree that the differences in T-Sapphire reporter output could be attributed to p65 oscillations. We had not previously considered this interesting possibility - which is not addressed by the present data - believe it is worth future investigation. As suggested by the reviewer, we have now omitted the TNF-a data, and agree that this change does not impact the overall claims of the paper.

2. Figure 3B, the authors introduced CARD9CARD-µNS as a stable condensed seed for BLC10. However, considering CARD9CARD can form polymers at high concentration (Figure 3B and S3D), are these high expression levels of CARD9CARD able to induce BCL10-mEos3.1 assembly (as measured by DamFRET in yeast cells)? Can the authors examine BCL10 FRET at these high expression level of CARD9CARD? We assume that BCL10 will be assembled in these cells. This would provide a valuable control experiment and support the author's conclusions.

Indeed, this question is amenable to DAmFRET. Accordingly, we have now performed DAmFRET of yeast cells expressing Bc10-mEos3.1 in the presence of either CARD9CARD-mCardinal or mCardinal itself (see new Figure S6A and B, and associated Results section). We confirmed that cells with high CARD9CARD-mCardinal expression had higher FRET on average than cells with low expression. Importantly, cells expressing high or low levels of mCardinal itself had the same FRET level (Figure S6).

3. Figure 3C, the text said "Whereas WT CARD9CARD assembled into polymers at high concentration, the pathogenic mutants R18W, R35Q, R57H, and G72S failed to do so (Figure 3C and S7B,C), explaining why they cannot nucleate BCL10". This claim that these mutants can not nucleate BCL10 does not have a figure call out or a reference. The authors then show the results in Figure 3E which supports this claim. Even though they were done in the context of full-length CARD, all proteins contain the I107E mutation that releases autoinhibition. For clarity, the authors should consider rearranging the text to avoid explaining a phenomenon and making conclusions before showing the results.

We have now rearranged this section to match the figures and claims.

4. Figure 4D, E and Video 1, the authors showed the nucleation of BCL10 into puncta within live cells is followed by p65 translocation to the nucleus. The authors claim that 'this result suggests that BCL10 is indeed supersaturated prior to stimulation' (paragraph 2 section titled BCL10 is endogenously supersaturated'). We fail to understand how this live-cell experiment leads to the conclusion BCL10 is supersaturated before stimulation. We think this text should be deleted from the text, or put into context with the DAmFRET data that lead the authors to make this claim. It would be interesting for the authors to define in discussion what are the golden criteria to claim a protein exists in a supersaturated state with live cells (by microscopy or other methods)? Adaptor protein assembly into puncta and the subsequent nuclear translocation of transcription factors is a common phenomenon across signalling pathways. Not all these pathways rely on signaling adaptors existing in a supersaturated state. The field of cell signaling (and cell biology in general) would benefit from a detailed definition of how these physical-chemical definitions of proteins are supported by experimental data. We believe that this paper will become a seminal paper in the field, and future work will benefit from a clear definition of how a claim of supersaturation is derived from the data.

We appreciate that the concept of supersaturation will be foreign to many biologists, and welcome this opportunity to elaborate. We have now rephrased the corresponding Results section for figure 4D, E, and have added new evidence to support our claim that BCL10 is supersaturated, as had been requested by reviewer 2 (see below in response to point 1). Supersaturation, as we (correctly) use the term, occurs when the concentration of a protein *in solution* exceeds its equilibrium solubility for the given conditions. The term is also sometimes used to describe *global* protein “concentrations” in excess of the solubility limit, even if a dense phase has already formed and potentially depleted the effective concentration (in solution) to the solubility limit. This is a key distinction, as only the former implies a high-energy out-of-equilibrium scenario that predetermines a future change -- release of the excess energy via phase separation.

How does one experimentally determine if a protein is supersaturated? In theory, one may conclude that a protein is supersaturated if its assembly causes a net loss of energy from the system (i.e. exothermic). Unfortunately, it is likely not yet possible to perform such measurements with sufficient sensitivity inside a living cell. However, it is possible to infer that a protein is supersaturated if assembly can be shown to occur without a net input of energy to the system, i.e. without any change in thermodynamic control parameters such as temperature, pH, post-translational modifications, concentration of the protein, or concentration of any interacting factor. To do this, one introduces a substoichiometric amount of pre-assembled protein to the system. This manipulation will trigger assembly if the protein is supersaturated. If the protein is instead subsaturated, assembly will not occur and the exogenously added assemblies will simply dissolve. This phenomenon, known as “seeding” in the prion field, is considered a golden criterion sufficient to conclude that a protein has prion behavior. However, because bona fide prions additionally require a means for dissemination between cells, seeding analyzed at the cellular rather than population level is more appropriately considered a sufficient criterion for supersaturation (which is a prerequisite for classical prion behavior (Khan et al. 2018)). Our CARD9CARD-Cry2 experiment was designed to test this criterion. Specifically, it allowed us to introduce a seed independently of receptor activation, thereby precluding any orthogonal cellular response that might lower Bcl10 solubility through e.g. a post-translational change. That the seeds were substoichiometric is evidenced by the fact that Bcl10 polymerized homotypically following stimulation (i.e. it didn’t just bind to the CARD9CARD puncta, but went on to deposit onto itself).

How does assembly under this scenario differ in principle from the many examples of puncta formed by other signaling proteins that occur upon stimulation of their respective pathways? Puncta formation that is induced by a thermodynamic change in the cell cannot be said to have resulted from pre-existing supersaturation. Rather, the stimulus may have caused some change that either increases the effective concentration of the protein (e.g. upregulates its expression, induces a post-translational change that activates it, or releases an inhibitory factor) or reduces solvent activity (e.g. change in pH).

An additional requirement (necessary but not sufficient) is that the assembly must be regular with respect to some order parameter. That is to say, it must be a bona fide “phase”. At a minimum, this implies a uniform density. Additionally, for supersaturation to persist over biological timescales under physiological conditions and confinement volumes, the assembly (once formed) must also have structural repetition in at least two dimensions, i.e. crystallinity (Rodríguez Gama et al. 2021; Zhang and Schmit 2016). We know this to be true for Bcl10.

Rodríguez Gama A, Miller T, Halfmann R. 2021. Mechanics of a molecular mousetrap-nucleation-limited innate immune signaling. Biophys J 120:1150–1160. doi:10.1016/j.bpj.2021.01.007

Khan, T., Kandola, T.S., Wu, J., Venkatesan, S., Ketter, E., Lange, J.J., Rodríguez Gama, A., Box, A., Unruh, J.R., Cook, M., et al. (2018). Quantifying nucleation in vivo reveals the physical basis of prion-like phase behavior. Mol. Cell 71, 155-168.e7.

Zhang L, Schmit JD. 2016. Pseudo-one-dimensional nucleation in dilute polymer solutions. Phys Rev E 93:060401. doi:10.1103/PhysRevE.93.060401

5. Regarding the supersaturated state of BCL10, the authors convincingly use optogenetics to show how transient assemblies of CARD-Cry2 can template BCL10 assembly. This is a convincing experiment that shows templated nucleation of BCL10. To strengthen the claim that BCL10 is supersaturated endogenously we suggest the author quantify the expression of BCL10-mScarlet and CARD-Cry2 and ideally show that this phenomenon can be observed at expression levels equivalent to endogenous.

As stated above, that BCL10-mScarlet formed polymers that we observed to elongate homotypically off of the CARD9CARD seeds indicates that the protein was supersaturated under the conditions of the experiment. The concentration of CARD9 is not a relevant parameter in this case. We had already compared the expression of BCL10-mScarlet to endogenous BCL10 in 293T, THP-1, and human fibroblast cells by quantitative immunodetection (Figure S10D), revealing that the expression level of our BCL10-mScarlet constructs matched that of endogenous BCL10, which was approximately the same in all cell lines. We also compared the distribution of expression levels of BCL10-mScarlet versus that of endogenous BCL10 using antibody staining followed by flow cytometry, which confirmed that the range of expression levels of BCL10-mScarlet falls within that of endogenous BCL10 in 293T cells (Figure S10F). Hence, we believe our data suffice to conclude that Bcl10 is supersaturated at endogenous levels of expression.

Minor comments:1. Special character "δ" is not displayed in the text (instead only a space).

This error occurred upon exporting the manuscript from our text editor to a PDF. We now have made sure all special characters are present in the PDF version.

2. Several cell lines including mouse, human, and yeast lines were used across this manuscript. It would be clearer and more helpful if the exact cell type of the line could be indicated. Such as, "BCL10-mEos3.1 yeast cells" instead of "BCL10-mEos3.1 cells", "BCL10-mScarlet HEK293T cells" instead of "BCL10-mScarlet cells".

We have now modified all instances to indicate the origin of the cell lines tested.

3. Figure 5B, the authors indicated that BCL10 colocalized with CARD9CARD, then please show the merged image as well.

We have now included the merged image to indicate colocalization in the inset images.

4. Figure 6E, authors claimed that cells were stimulated with blue light for the indicated durations. The longest duration is 12 hours. Please specify if it was continuous exposure or several rounds of exposure in the indicated durations.

We have now specified in the figure legends, text, and methods section, that this specific experiment used a continuous exposure of blue light.

Reviewer #1 (Significance (Required)):This work used a combination of FRET and optogenetic tools to engineer CBM signaling and visualize the effects. They incorporated knowledge from structure biology, together with their results from mutations and truncations, dissected the significance of each protein in CBM signalosome, and demonstrated in detail how higher-order assemblies make all-or-none cellular decisions. We believe this paper will be a seminal paper in the field of cell signalling and cytoplasmic organization. It defines a new paradigm of macromolecules assembly of signalling complexes as being dependent on protein existing in a supersaturated state. Importantly this paper opens up new questions regarding macromolecular signaling complexes (found in many innate immune signaling pathways): How is protein supersaturation maintained and used throughout evolution to construct biochemical signalling switches?This paper will be of particular interest to scientists working on immunity and cell signalling, especially in the field of higher-order assemblies. However, we feel the impact of this paper goes beyond these fields, and we believe this manuscript will be of broad interest to the cell biology and biophysics communities. For reference, our expertise is in innate immunity and cell biology.Reviewer #2 (Evidence, reproducibility and clarity (Required)):In their manuscript entitled "A nucleation barrier springloads…" Rodriguez-Gama et al. dissect the assembly mechanism of the signalosome, composed of the proteins CARD9, BCL10 and MALT1, using a novel in-cell biophysical approach (DAmFRET). They first overexpressed fluorescently tagged versions of the proteins to promote their assembly in yeast and mammalian cells, finding that CARD9 forms higher order assemblies across a wide range of concentrations with no discontinuity in the DAmFRET profile. In contrast, the DAmFRET profile of BCL10 showed a clear separation between monomers and higher order assemblies, which started to form spontaneously only at higher BCL10 concentrations. Furthermore, at the two states of the proteins co-exist at all concentrations. These observations imply that there is a nucleation barrier to forming BCL10 assemblies. MALT1 showed no change in FRET regardless of its expression level. These observations, alongside fluorescence microscopy of the assemblies, and previous structural studies, suggest that BCL10 forms self-templating polymers that act as a switch for an all-or-nothing immune response, assayed in this case by monitoring the nuclear translocation of the NF-κB subunit p65. The authors also assessed the effects of known disease-causing mutations on the nucleation barrier, showing that changes in the strength of the nucleation barrier can have major effects on signalosome function. Finally, they used optogenetic methods to trigger assembly of individual signalosome components, providing insight into the minimal components/conditions required for signalosomes to work.Major commentsOverall, the experiments by Rodriguez-Gama et al. offer convincing evidence that there is a nucleation barrier to BCL10 polymerisation, and that a CARD9 template is sufficient to overcome the barrier. Although the existence of a nucleation barrier had already been postulated, based on structural and other studies (referenced by the authors), it had lacked a rigorous demonstration. This work provides that demonstration, which is important for the signalosome field and more broadly applicable to researchers studying cellular decision making. The study further demonstrates that DaMFRET is an excellent to study protein assembly processes in their native environment, allowing the authors to tackle a question that would have been technically very difficult to address otherwise. The optogenetic experiments are a nice sufficiency test for their ideas.We feel there are a few key points to address before publication.1) One of the main conclusions is that spring-loading the nucleation barrier with high super-saturating BCL10 concentrations allows a decisive response. Although much of the data strongly imply this conclusion, the dependence of the immune response on BCL10 concentration was not tested directly. A key prediction of the nucleation barrier is that at concentrations below saturation, BCL10 should not be able to induce an all-or-nothing response when stimulated. At saturated/super-saturated concentrations BCL10 should be able to induce a response. At deeply super-saturated concentrations the response should start to be activated spontaneously in the absence of an external stimulus. These predictions could be tested using the doxycycline-inducible BCL10 system (Figure S2D), without establishing major new experimental avenues. We feel that such an experiment would strengthen the main conclusion. It might also help to shed light on whether being highly supersaturated enables a more decisive response than being just saturated.

This is a great idea. As the reviewer suggested, our Doxycycline-inducible BCL10 system enables us to induce and track the state of BCL10 over time. We have now performed the requested experiments (Figure S9D, E) and incorporated the results into the relevant section of the text. In short, our new analyses show that BCL10 indeed has a concentration threshold for activation by stimulation, and that it can also nucleate spontaneously when overexpressed. Note that our original analyses in Figure 4B and C also demonstrate spontaneous BCL10 activation at high concentrations. With this new evidence and the orthogonal approaches used in Figure 5, we believe our data definitively support our conclusion that BCL10 is supersaturated.

2) Intuitively, readers might expect that if BCL10 is supersaturated then, once nucleated, it would rapidly assemble at the nucleation sites. In Figure 5B, CARD9CARD-miRFP670nano-Cry2 assemblies are optically induced throughout the cell. However, BCL10 appears to nucleate at just a few sites with a few minutes delay. More widespread nucleation and growth of BCL10 polymers seems to take longer (20-40 minutes, Figures 5B and 5C), after CARD9CARD-miRFP670nano-Cry2 has disassembled. Furthermore, in Figures 4D and 4E, very few BCL10 assemblies are visible/quantifiable after 70 minutes PMA exposure, but p65 has clearly entered the nucleus. It looks like BCL10 assembly slightly lags behind p65 nuclear entry. Can the authors provide a more detailed explanation of these kinetics?

We do note that the number of CARD9CARD clusters formed upon opto-stimulation exceeds the apparent number of BCL10 nucleation sites. We believe this is consistent with nucleation-limited kinetics, where the clustering of CARD9-CARD increases the local probability of nucleation. As nuclei form and grow, they lower the probability of subsequent nucleation elsewhere in the cell. Additionally, it is possible that our artificial seeds do not perfectly mimic the native CARD9 seeds that form upon natural stimulation (e.g. due to potential steric interference from the fluorophore and Cry2). We also acknowledge that there is a slight delay in the visible appearance of BCL10 polymers relative to p65 nuclear translocation. We expect that MALT1 activates already when the polymers are still too small to see (sub-resolution), whereas the polymers only become microscopically visible once they’ve grown quite a bit more.

3) Related to point 2 above, in Figure 5D, the leftmost cell in the field of view clearly contains CARD9CARD assemblies but there are no BCL10 assemblies and p65 is not imported into the nucleus (in contrast to the central cell in the field of view). How often does CARD9CARD optogenetic assembly lead to BCL10 assembly? In other words, can the authors quantify the cell-to-cell variability in this experiment?

Throughout our experiments, whether analyzing BCL10 puncta formation, NF-κB transcriptional activity, or p65 translocation, we observed a persistent nonresponsive fraction of cells even at saturating levels of stimulation. Specifically, approximately 30% of THP-1 cells failed to acquire T-Sapphire fluorescence or form BCL10-mEos3.2 puncta when stimulated with high levels of β-glucan (Figure 1B and E, respectively), and approximately 25% of 293T cells failed to acquire T-Sapphire fluorescence or exhibit p65 nuclear translocation when stimulated with high levels of PMA (Figure 1C and Figure 4E, respectively). Because these numbers did not depend on whether BCL10 was endogenously or exogenously expressed, we know that the underlying cell-to-cell heterogeneity involves factors upstream of BCL10. Indeed, the fraction of recalcitrant cells drops to 10% in our optogenetic experiments that bypass upstream factors (Figure S11E). Possible sources of heterogeneity include different physiological states of the cells or fluctuations in the expression levels of any upstream factor in the signaling pathway. We believe that this phenomenon is not unique to the CBM signalosome, as we (unpublished) and others (Fernandes-Alnemri T et al., 2009, Dick M et al., 2016) have similarly observed a fraction of non-responding cells upon activation of the inflammasome, which involves nucleation-limited polymerization of the adaptor protein ASC. While this phenomenon is interesting and may be important to our understanding of the full complexity of signalosomes in vivo, we believe that identifying the source of heterogeneity would be outside the scope of the present manuscript. We now describe this phenomenon in the final paragraph of the “Endogenous BCL10 is constitutively supersaturated” section.

Fernandes-Alnemri, T., Yu, JW., Datta, P. et al. AIM2 activates the inflammasome and cell death in response to cytoplasmic DNA. Nature 458, 509–513 (2009). https://doi.org/10.1038/nature07710

Dick, M., Sborgi, L., Rühl, S. et al. ASC filament formation serves as a signal amplification mechanism for inflammasomes. Nat Commun 7, 11929 (2016). https://doi.org/10.1038/ncomms11929

Minor commentsWhile the work is scientifically well done, the text reads as though it is meant for experts rather than a broad audience. This is a pity because it risks alienating readers. We suggest that some adjustments to the text (mainly additional explanations and not ruling out alternative interpretations of the data) would widen the audience and increase the impact of this important study. Below are some suggestions that might help.1. In the first Results section, the authors write: 'This suggests that Bcl10 but not CARD9 assembly occurs in a highly cooperative fashion that could, in principle (Koch, 2020), underlie the feed forward mechanism.' It isn't obvious how Figure 1 leads to this statement. Could the authors give a more detailed explanation?

We have now revised the text to elaborate on this interpretation.

2. One limitation of DAmFRET is that it can only detect a nucleation barrier where there is a difference in FRET between the monomer and the assembled form of the protein. However, it can't necessarily detect when there is not a nucleation barrier i.e. if there's no difference in FRET. The text seems to suggest that CARD9 and MALT1 don't have nucleation barriers to their assembly. While this might not be intentional, it would be helpful to explicitly state that CARD9 and MALT1 could also possess such barriers that are not detectable by this method. This wouldn't detract from the finding that BCL10 has a barrier that plays an important function.

The reviewer is correct that DAmFRET would not be able to detect a nucleation barrier if the assembled phase does not condense the fluorophore to a sufficiently high density for FRET to occur. In our experience, this is only a concern for very large proteins whose bulk “dilutes” the fluorophores within the assembly. Death domains, on the other hand, are only ~ 3 nm in diameter, and FRET occurs within a range of ~10 nm; hence we think it very unlikely that the death domains could be forming cryptic polymers that escape our detection. In any case, when assembly does produce a change in FRET, we can with confidence determine how strongly that form of assembly is governed by concentration. Hence, for CARD9, which does produce a FRET signal upon assembly, we can say that assembly has a smaller intrinsic nucleation barrier than that of BCL10. We further eliminated the possibility of multi-step nucleation (which would reduce the apparent nucleation barrier relative to the one-step ideal case) for CARD9 by showing that artificial condensates of the protein expressed in trans do not influence the concentration-dependence of FRET (Figure 4 B). Finally, under all conditions where CARD9 lacked FRET, it also lacked signaling activity, suggesting there is not a cryptic functional assembly that evades our assay. Likewise MALT1, which lacked FRET at all concentrations, was entirely unable to activate NF-κB upon overexpression (Figure S8 A and B), suggesting that it too is not forming a cryptic functional assembly that evades our assay. We therefore feel confident in our conclusion that CARD9 and MALT1 lack nucleation barriers of a magnitude comparable to that of BCL10. Note that our claim is not that they entirely lack a nucleation barrier (CARD9 after all does form a multi-dimensionally ordered polymer), but rather that we fail to observe a nucleation barrier and hence any barrier that may exist is insufficient to manifest at the cellular level.

3. In the final Results section, the idea that MALT1 activation doesn't depend on BCL10 polymer structure doesn't necessarily follow from the data. An alternative interpretation is that optogenetic clustering of MALT1 causes it to recruit BCL10 and form BCL10-MALT1 filaments (structure solved by Schlauderer et al., 2018). Also, the optogenetic clustering of MALT1 may mimic some structure found in the BCL10 cluster. Therefore, we are neither convinced that the data unambiguously show that MALT1 activation strictly depends on multi-valency rather than an ordered structure of BCL10 polymers nor that this conclusion is truly necessary for the paper.

We agree that the reviewer’s alternative interpretation of this experiment is possible. However, we consider it unlikely because we performed the experiment with MALT1 lacking its Death Domain (residues 126-824), which mediates its interaction with BCL10 (Schlauderer et al., 2018). Our experiments then suggest that MALT1 clustering is sufficient for activation independent of any structuring mediated by BCL10. Nevertheless, we have now performed an additional control in which we treated these cells with PMA to induce BCL10 polymerization. As expected, the NF-κB transcriptional reporter utterly failed to activate in this condition, indicating that MALT1 does not interact with BCL10 polymers when it lacks its death domain. This aspect has been further elaborated in our response to reviewer 3 point 5.

4. What optical density do the yeast cells reach during the 16h induction in galactose? If they are in stationary phase, this could affect the assembly status of the proteins being expressed, as the cytoplasm becomes glassy when cells are starved, and this coincides with widespread protein aggregation/assembly (Joyner et al., 2016; Munder et al., 2016).

In our DAmFRET strategy, we first dilute an overnight culture and regrow the cells to log phase prior to resuspending them in galactose media. Our strain is engineered to undergo cell cycle arrest upon protein induction, hence exponential growth is prevented and the cells do not deplete galactose during the 16 hr induction. We have also performed many time courses of DAmFRET following induction and generally find no qualitative difference between early and late times (unpublished). Early time points simply have lower expression and correspondingly fewer cells in the high FRET state. Importantly, all comparisons between proteins are made with the same 16 hr induction.

5. Although these experiments show that thermodynamically lowering the BCL10 nucleation barrier (e.g. by post-translational modifications or protein expression levels) isn't required for a response, they don't rule it out. It would be good to state this in the discussion, as cells may have multiple mechanisms of switching on the signalosome.

We thank the reviewer for this suggestion and have now explicitly stated in the discussion that our experiments do not argue against possible thermodynamic tuning of the nucleation barrier.

6. The discussion compares signalosomes with condensates formed by liquid-liquid phase separation. This is an interesting comparison but it suggests that disordered assemblies would not be capable of performing signalosome-like functions. This needs to be explained more clearly. For example, non-amyloid prions seem to form gel-like assemblies with a high nucleation barrier that are capable of driving heritable traits, likely through self-templating (Chakravarty et al., 2020). Such examples could represent disordered assemblies with signalosome switch-like behaviour. Furthermore, there are examples of condensates that are induced by environmental changes e.g. Pab1 and Ded1 condensates (Riback et al., 2017; Iserman et al., 2020). This potentially allows the proteins to reach high concentrations and remain un-condensed until a change in heat or pH overcomes a nucleation barrier required for condensate formation. Although the condensates aren't self-templating, they seem to require energy for their disassembly. Combined, this also allows switch-like behaviour, where the switch is flipped back to the uncondensed off state once conditions return to normal. In general, crossing a phase boundary can represent a switch-like response. Finally, recent electron-tomography experiments show that ASC puncta comprise clusters of filaments (Liu et al., 2021, biorxiv). CARD9/BCL10 assemblies may have similar ultrastructures and liquid-liquid phase separation may well play a role in their assembly.

Indeed, we explicitly maintain that liquid phases cannot themselves perform signalosome-like functions. Chakravarty et al. 2020 did not observe amyloids associated with their phenomena, but the relevant experiments were not designed to exhaustively exclude an underlying ordered phase. To the extent that gelation is involved, their observations are fully consistent with ours. IUPAC defines a “gel” as a colloidal network involving a solid phase and a dispersed phase. The existence of a solid phase necessarily implies an underlying disorder-to-order transition, even if limited to small length scales. In the case of gelation associated with liquid-liquid phase separation, nucleation of the ordered phase simply occurs in two steps (first condensation, then ordering). Note also that a liquid phase could in principle give rise to a heritable phenotype if it activates a positive feedback in a molecular biological process involving the protein of interest (e.g. upregulation of its expression or a change in interacting factors). Chakravarty et al. did not exclude such phenomena (it would be very difficult to do so); hence it cannot be concluded that phase separation is responsible for the sustained phenotypic changes.

We do not fully follow the reviewer’s logic concerning the relevance of Pab1 and Ded1 condensates. These proteins only condense when their respective phase boundaries fall below the endogenous protein concentration, as upon thermal stress. The proteins are not supersaturated in the absence of such conditions (for example, they cannot be seeded), and it is incorrect to characterize the change in heat or pH as overcoming a pre-existing nucleation barrier. The concept of a nucleation barrier only applies under conditions where a phase is thermodynamically favored. It is also misleading to state that the Ded1 and Pab1 condensates require energy for disassembly. Rather, they require energy to disassemble *rapidly*. Unless the assemblies have accessed a more ordered phase as described above (two step nucleation), involving a lower phase boundary, they will inevitably dissolve after the conditions return to normal.

We have much prior experience with ASC. Although it has not been explicitly shown, that it forms ordered polymers and can behave as a prionoid in vivo suggests that it very likely operates the same way as BCL10 (i.e. is physiologically supersaturated). That full-length ASC forms clusters of filaments is not relevant (in our view) to the mechanism shown here, which only requires that filaments are indeed formed. Formally, the size of the relevant nucleus determines the minimum length scale at which ordering must manifest in our mechanism. Based on the structure of death domain filaments, this could be as small as tetramers or hexamers (a minimal but structurally complete “polymer”).

As stated above, and now elaborated in the discussion, our data do not exclude a role of thermodynamic regulation, as could lead to liquid-liquid phase separation, in tuning the nucleation barrier of Bcl10. What they do exclude is that such changes are required for Bcl10 to activate in the first place.

7. Can the authors comment on the loss of BCL10 in Echinodermata, Anthropoda, Nematoda? Is there another protein that plays a similar role? Could a CARD or PCASP protein possess self-templating properties? Could other methods of control be at play e.g. protein expression?

This is a very interesting question! We think the reviewer’s suggested explanations for the loss of BCL10 in those lineages are valid and worthy of future exploration. Nematodes such as *C. elegans* have lost multiple components of innate immunity. They have very few pathogen recognition receptors and also lack NF-κB! They do, however, have other adaptor proteins that the literature and our unpublished data suggest may have self-templating ability, such as TIR-1. *Drosophila* also encodes multiple TIR-containing proteins that are essential for innate immunity. In short, it is possible that other proteins have acquired the hypothetically essential role of supersaturation and nucleation-limited signaling in these organisms.

8. Figures 1B/1C: Can the authors comment on why the active cells plateau at about 70-75%? This is a striking feature of the plots, but the explanation may not be obvious to readers.

See our response to major point 3, above.

9. Figures 1D/1E: What was the concentration of B-glucan used in this experiment? This could be included in the figure legend. If greater than 1ug/ml this means that the % of active cells in Figure 1B matches the % of cells with BCL10 assemblies in Figures 1D/1E, which is potentially an important point.

We thank the reviewer for bringing this point to our attention. We have now indicated in the figure legend the concentration of B-glucan used in this experiment (10 μg/ml). That the percentage of active cells in Figure 1B matches that of cells containing BCL10 polymers in Figure 1D and E indeed strengthens the stated relationship between BCL10 assembly and NF-κB activation in THP-1 cells subjected to a relatively physiological stimulus. Additionally, we have performed experiments to measure the levels of p65 translocation in THP-1 cells treated with B-glucan that express BCL10-mEos3.2. This data is shown in Figures S1D and E in response to reviewer 3.

10. Use of both 'BCL10' and 'Bcl10' when referring to the protein.

We have now replaced all instances where Bcl10 was used to follow guidelines for gene and protein name conventions.

Bruford EA, Braschi B, Denny P, Jones TEM, Seal RL, Tweedie S. Guidelines for human gene nomenclature. *Nat Genet*. 2020;52(8):754-758. doi:10.1038/s41588-020-0669-3

11. In the supplementary figures there are some formatting problems/missing words in the figure legends. In Figure S11 there is a black box covering the lower part of the figure.

We have now fixed these instances.

References used in this reviewChakravarty, A.K. et al. (2020) "A Non-amyloid Prion Particle that Activates a Heritable Gene Expression Program," Molecular Cell, 77(2), pp. 251-265.e9. doi:10.1016/j.molcel.2019.10.028.Iserman, C. et al. (2020) "Condensation of Ded1p Promotes a Translational Switch from Housekeeping to Stress Protein Production," Cell, 181, pp. 818-831.e19. doi:10.1016/j.cell.2020.04.009.Joyner, R.P. et al. (2016) "A glucose-starvation response regulates the diffusion of macromolecules," eLife, 5. doi:10.7554/eLife.09376.Munder, M.C. et al. (2016) "A pH-driven transition of the cytoplasm from a fluid- to a solid-like state promotes entry into dormancy," eLife, 5(MARCH2016). doi:10.7554/eLife.09347.Riback, J.A. et al. (2017) "Stress-Triggered Phase Separation Is an Adaptive, Evolutionarily Tuned Response," Cell, 168(6), pp. 1028-1040.e19. doi:10.1016/j.cell.2017.02.027.Schlauderer, F. et al. (2018) "Molecular architecture and regulation of BCL10-MALT1 filaments," Nature Communications 2018 9:1, 9(1), pp. 1-12. doi:10.1038/s41467-018-06573-8.Reviewer #2 (Significance (Required)):The existence of a nucleation barrier had already been postulated, based on structural and other studies (referenced by the authors), it had lacked a rigorous demonstration. This work provides that demonstration, which is important for the signalosome field and more broadly applicable to researchers studying cellular decision making. The study further demonstrates that DaMFRET is an excellent to study protein assembly processes in their native environment, allowing the authors to tackle a question that would have been technically very difficult to address otherwise.Reviewer #3 (Evidence, reproducibility and clarity (Required)):The study by Rodriguez Gama et al. addresses the molecular function of CBM complex-forming proteins CARD9, BCL10 and MALT1 in the activation of myeloid cells, using optogenetic tools, transcriptional reporters and biochemical approaches. It is known from previous studies that Bcl10 oligomerizes into filamentous oligomeric structures incorporating Malt1, and that these structures are nucleated by receptor-induced activation of CARD proteins such as CARD11 (in lymphocytes) or CARD9 (in myeloid cells), but the mechanism underlying the assembly of the resulting CBM complexes remain incompletely understood.The authors develop beautiful optogenetic tools to address this question, and convincingly demonstrate that CARD9-mediated nucleation of BCL10 triggers a binary cellular NF-κB response in a spring-load-like fashion, and identify mutants of BCL10 and CARD9 that impact this capacity. Unfortunately, however, the authors do not do a good job to simplify this complex problem so it can be easily understood. In particular, the choices of mutants, models and experiments are not consistent between figures, and some data seem to be arbitrarily added or omitted. Complex hybrid constructs are also used, without assessing whether these are indeed functional in the corresponding ko cells. The paper would therefore benefit from a major overhaul. We also noticed that the literature is often not cited adequately and have included a (non-exhaustive) list of examples of wrong, incomplete, or erroneous citations below.1. The initial observations of binary signaling are derived from a reporter system. Although there are controls to show that the reporter used does not function intrinsically cooperatively, it would be nice to see additional data to show that cooperativity occurs also at the level of endogenous response systems, for instance by qPCR-based assessment of a natural NF-κB target gene (induced for example by TNFa versus B-glucan in THP-1 cells, and by TNFa versus PMA in 293T cells).

As detailed in the introduction, NF-κB has been shown by multiple labs to activate in a binary fashion. Our manuscript shows that NF-κB activation occurs in a binary fashion both at the level of transcription and at the level of nuclear translocation (upstream of any transcriptional output). While we do agree that additional data could further illustrate the biological significance of our findings, we do not feel it is necessary for our conclusions. Note also that because NF-κB activation occurs in a binary fashion per cell, a simple qPCR experiment would not suffice to extend our findings to the broader NF-κB regulon. Instead, one would have to use e.g. RNA-FISH or single cell RNA-seq, nontrivial experiments that would take months to complete.

2. The cell lines in Figures 1D-E (and also some of the BCL10 mutants used later on) would have been better run in the assays in the early parts of Figure 1. The final conclusion prior to the section The adaptor protein BCL10 is a nucleation-mediated switch is otherwise not justified. This is a central tenet of the paper, that is referred to again, with some other ancillary data to support it. These mutants reappear later in the paper, but it would have been better, and easier to make rescue lines of BCL10 KO in Figure 1, otherwise the logic is lost, and the models seem chosen arbitrarily.

The choice of experiments in different panels of Figure 1 resulted from a chronological progression of reagent construction as the project evolved. We do appreciate that switching between the assays may lead readers to doubt one or the other. Therefore, we have now immunostained for endogenous p65 in the same experiment as for Figure 1D and confirmed that p65 translocated to the nucleus only in THP-1 BCL10-KO cells that have been reconstituted with WT BCL10-mEos3.2, but not E53R. We think this additional evidence along with our orthogonal measurements in other reporter systems confirms our findings that BCL10 nucleation determines NF-κB activity.

3. Expression with microNS is not well controlled and gives little real evidence for what is occurring. It is unclear what the concentration of the protein expressed was, but certainly the relative expression of the CARD9(CARD) and the microNS version should be assessed.

We believe these concerns result from a misunderstanding. We assume the reviewer is referring to the experiment in Figure 3B. Expression of muNS on its own has no effect on the DAmFRET of other proteins, and we have previously used it in exactly the same way as here (Holliday M et al. 2019 and Kandola T et al. 2021). Please note that muNS fusion proteins in our experiment have an orthogonal fluorescent protein whose spectra do not significantly overlap with those of mEos3.1. The experiment evaluates a protein’s ability, when condensed via its fusion to muNS, to nucleate an mEos3.1-fused protein that is expressed in trans. Fusion of proteins to muNS does not affect their expression levels, as we now show for CARD9CARD-muNS-mCardinal versus CARD9CARD-mCardinal (Figure S6D).

Also, the AmFRET profile of CARD9CARD looks very weird, it cannot be compared to BCL10.

We are unsure in what way the AmFRET profile of CARD9CARD is “weird”. It is fully consistent with expectations and has been thoroughly explained in the text. We suspect the reviewer was bothered by the sharp acquisition of FRET at approximately 100 uM. As explained in the text, this represents the phase boundary, also known as the solubility line, for CARD9CARD polymers, which we previously showed in vitro (Holliday M et al. 2019). Above this concentration, the protein self-assembles without a nucleation barrier, hence the sharp but continuous change in FRET. BCL10 plots, in contrast, show a discontinuous acquisition of FRET, which indicates a nucleation barrier. In order to highlight that the CARD9CARD transition is understood and expected, we have also now added a line to the plot to demarcate the phase boundary.

4. We are not convinced of the usefulness of the introduction of a slew of disease-causing CARD9 mutations that may or may not be relevant to the authors' point. The fact that they do or do not function in a specific sub portion of an assay that may or may not be relevant to biological activity seems to be of interest but without biochemical understanding, little is clear.

While several reports have shown the clinical importance of these CARD9 mutations on susceptibility to fungal infections, little was known about the molecular mechanism underlying their effects. The inclusion of the disease-causing mutants to this paper is justified for the following reasons. First, they demonstrate the relevance of our work to disease. Second, they build off our findings to provide an otherwise unknown molecular mechanism of these mutants. We showed using independent methods that CARD9CARD mutations disrupt the ability to nucleate BCL10, via two different mechanisms. Finally, validating the disease-causing mutations allowed us to use them as controls for subsequent experiments demonstrating that BCL10 is supersaturated.

5. The Optogenetic experiments are interesting, but difficult to interpret without evidence that these MALT1 constructs are indeed still functional when expressed in MALT1-deficient THP-1 cells. We do not therefore think that this experiment shows a necessity for clustering to signal, just a sufficiency, and in a highly artificial construct.

We welcome the opportunity to elaborate on the optogenetic experiments. Since BCL10 and MALT1 are expressed ubiquitously across cell types, the validity of our findings should not depend on the cell type used. Indeed, much of what we already know about innate immunity signalosomes comes from work in HEK293T cells. Our optogenetic experiments using MALT1 were performed in 293T MALT1-KO cells in Figures 6E and F, and employed two distinct functional assays (p65 nuclear translocation and a transcriptional reporter). While our approach employs light to control clustering, similar approaches using (no less-artificial) chemically induced dimerization domains have been used to study caspase activation (Oberst A et al., 2010, Boucher D et al., 2018). Our use of light affords higher specificity, reversibility, and spatial and temporal control over MALT1 assembly than does chemically induced dimerization.

To demonstrate the necessity of clustering, we have now performed an experiment with MALT1(126-824)-miRFP670-Cry2 expressed in 293T MALT1 KO cells that contain a transcriptional reporter of NF-κB ,as in figures 6E and F. We added PMA to the cells and found that it failed to activate NF-κB (Figure 6), confirming that the interaction of MALT1 (via its death domain) with polymerized BCL10 is required for activation. Note that MALT1 and BCL10 exist as a soluble heterodimer prior to BCL10 polymerization; hence it is polymerization, rather than the interaction itself, that activates MALT1. That artificial clustering rescues this defect strongly suggests that the effect of polymerization can be attributed to increased proximity rather than some allosteric effect communicated from BCL10 polymers through the MALT1 DD to its caspase-like domain.

Oberst, A., Pop, C., Tremblay, A.G., Blais, V., Denault, J.-B., Salvesen, G.S., and Green, D.R. (2010). Inducible dimerization and inducible cleavage reveal a requirement for both processes in caspase-8 activation. J. Biol. Chem. *285*, 16632–16642.

Boucher, D., Monteleone, M., Coll, R.C., Chen, K.W., Ross, C.M., Teo, J.L., Gomez, G.A., Holley, C.L., Bierschenk, D., Stacey, K.J., et al. (2018). Caspase-1 self-cleavage is an intrinsic mechanism to terminate inflammasome activity. J. Exp. Med. 215, 827–840.

6. In the introduction and other parts of the paper, there are numerous instances where the previous literature in the field is not adequately cited. Examples include:- In the introduction, it is weird to cite one original paper (a MALT1 ko study by Ruland et al., 2001; there are several other studies of ko papers for CBM components that would merit being citated along with this study) together with two reviews on that topic (Ruland and Hartjes 2019 and Gehring et al. 2018)- In the introduction, the original study by Wang et al., 2002 should be cited together with Rebeaud et al., 2002; the two studies on the same topic were published back-to-back- In the introduction, the statement "CARD10 and CARD14 are expressed in nonhematopoietic cells including intestinal and skin epithelia, respectively" should be supported by citations.- Still in the introduction, the 2 references for the statement "… CARD14 gain of function mutations cause psoriasis (Howes et al., 2016; Jordan et al., 2012)" are not appropriate. There are several reports of patients with CARD14 mutations (the study by Jordan et al. is only one of them) and several CARD14 mouse models that provoke a psoriasis-like phenotype, which would merit being cited.- In the following sentence: "Point mutations and translocations involving BCL10 and MALT1 cause immunodeficiencies (Ruland and Hartjes, 2019), testicular cancer (Kuper-Hommel et al., 2013), and lymphomas (Zhang et al., 1999).", the citation style also seems completely random, combining the citation of a single original paper for lymphomas (Zhang et al. 1999) (there are several other important original studies on that topic or recent reviews that could be cited instead), together with a review on immunodeficiencies (Ruland and Hartjes, 2019) and then another single example for a role of BCL10 and MALT1 in carcinoma (the study by Kuper-Hommel et al. is one, but several other original publications exist on the latter topic, showing for example a role in breast carcinoma or glioblastoma).- In the first section of the results, the reference cited for endogenous CARD10 expression in 293T cells (Ruland et al., 2001) is wrong, no endogenous CARD10 expression was assessed in that study

We have now revised the citations mentioned above and other instances to ensure adequate citations in each case.

Reviewer #3 (Significance (Required)):The paper deals with a complex question, namely how the CBM signalosome assembles and functions to stimulate NF-κB signaling. This question is important to the understanding of pro-inflammatory immune responses and basic life sciences in general. As the focal point of the paper is complex, and tools to study such phenomena are at the limit of technical capabilities, this further increases the potential impact of the work.Reviewer #4 (Evidence, reproducibility and clarity (Required)):The characterization of open-ended signalosomes in a number of innate-immunity and cell-death pathways, in particular formed by domains from the death-fold family, has led to the suggestions that these complexes allow a switch-like signalling response suitable for these pathways. It appears that this has been widely accepted. However, these suggestions are based largely on indirect observations and speculation.Rodriguez-Gama and coworkers have decided to test these suggestions more directly. Their results confirm the suggestions. Based on my own experience, papers that validate widely adopted suggestions are often not considered seriously by top journals, who are looking for hot topics/paradigm-changing/surprising type results. I would urge the editors to consider seriously work such as in this paper, which directly tests important suggestions and does so at a technically high standard. The authors use a range of ingenious approaches, both with recombinant proteins and in cells, and including proteins from organisms from different parts of the evolutionary tree, to support their interpretations, so it is an extensive and high-quality study. I am impressed that so many different fusion proteins with fluorescent tags continued to function as expected, but I guess the authors controlled for this as much as they could.Having said all this, I do get the feeling the authors are "over-selling" the nucleation barrier aspect of these signalling mechanisms. It is clearly an important and critical aspect of signalling in many systems, but then it is not the only important aspect; a number of other regulatory inputs play a role in different systems. So the statement "Our findings introduce a novel structure-function paradigm" in my view is overstretching things somewhat. Further in the Discussion section, the authors state "Existing explanations for the preponderance of ordered polymers in immune cell signalosomes have centered on the functions of multivalency at steady state, such as scaffolding and sensitivity enhancement resulting from the cooperativity of homo-oligomerization". They cite a small (and non-exhaustive) number of papers discussing this topic; all these include "seeding" or "nucleation" as an important part of the proposed mechanism. So I suggest the authors provide a more balanced discussion of this aspect. Different pathways appear to display a different level of switch-like behaviour, and one thing that the current version of the manuscript is missing is more discussion of other death fold-based systems and how the results on the CBM signalosome apply to these, and also other systems such as TIR domain-based ones, which currently get no mention whatsoever. In the CBM system, there seems to be one main nucleation barrier; can there be more than one in others?

We appreciate the reviewer’s perspective and have now acknowledged in the introduction and discussion additional prior literature that has paved the way for our study. Nevertheless, we maintain -- as now stated in the abstract -- that “our results defy the usual protein structure/function paradigm, and demonstrate that protein structure can evolve via selection for energetic maxima in addition to minima”. We have elaborated in the introduction and discussion how immune signaling provides the functional context in which such a paradigm can evolve, and how our findings uniquely support the paradigm.

One other aspect I need to express some criticism about is attention to detail – especially with a paper focusing on the physics behind biological processes, I would expect a higher standard of getting the terminology and units correct – see specific examples below. This can obviously be fixed easily.Specific points are listed below. No page or line numbers are provided so I have done my best to make it clear what the comments refer to.1. Abstract line 6 and throughout: in "NF-κB", the "k" is supposed to be "kappa" (Greek letter) – it stands for "nuclear factor kappa-light-chain-enhancer of activated B cells", not fully defined in the manuscript as far as I can see. Occasionally, small k is also used instead of the small cap K or whatever the authors used most of the time, but I don't think any of them use the Greek letter.

We had indeed used a version of the small “kappa” κ. We have now fixed the cases where we mistakenly used k instead of κ.

2. Page 2 (Introduction) paragraph 2 line 9: period missing at the end of sentence. Same Page 4 (Results: Assembly) paragraph 4 line 3.

This is now fixed.

3. Page 2 (Introduction) paragraph 2 line 15 and throughout: in long sentences, more commas can help help readability, for example before "leading" here. Similar page 15 paragraph 2 line 3 after "Additionally", paragraph 4 line 2 before "which".

We have now included more commas and tried to improve readability throughout.

4. Page 4 (Results: Assembly) paragraph 2 line 2: is "positive feedback" different from "cooperativity"? Is it a broader term that includes cooperativity, nucleation and other mechanisms? It may be useful to introduce some of these terms to avoid confusion by the readers.

“Positive feedback” is the broadest term as it is agnostic to mechanism. “Nucleation” refers to the initiation of a first order phase transition, which is one mechanism of positive feedback. Nucleation involves “cooperativity”, in that a higher order species is more stable than smaller species. However, cooperativity can occur for oligomers of finite size, whereas nucleation is reserved for phase transitions to species of infinite size. We appreciate that the use of so many related terms may have created more confusion than necessary. Hence, we have now revised the text to omit the more general terms -- “positive feedback” and “cooperativity” where possible.

5. Page 4 (Results: Assembly) paragraph 2 line 3: please define "TNF".

We have now fixed this and other acronyms.

6. Page 4 (Results: Assembly) paragraph 3 line 2: the use of size-exclusion chromatography to follow the size of complexes would assume that they are irreversible or very stable. It appears this may be the case here, but some discussion may be warranted.

We have now explained that SEC is appropriate for this experiment because large nucleation barriers generally imply stable assemblies.

7. Page 4 (Results: Assembly) paragraph 3 line 4 and throughout: the symbol for "kilodalton" is "kDa".

We have now fixed this mistake.

8. Page 4 (Results: Assembly) paragraph 3: I am not sure how the results discussed in this paragraph demonstrate that assembly occurs in cooperative fashion – just that there is a change in oligomeric states upon stimulation.

Cooperativity is implied by the absence of oligomer sizes between monomer and the large assembly. Nevertheless, we realized this can only be concluded in the case of homotypic assembly, which we cannot yet assume at this point in the paper. Therefore, we have revised this paragraph to say that the distribution is “consistent with” an underlying phase transition (which we then go on to prove).

9. Page 4 (Results: Assembly) paragraph 4 line 2: "WT" is not defined. Wild-type what? I presume "protein"?

We refer here to the wild-type protein. We have now fixed this mistake.

10. Page 4 (Results: Assembly) paragraph 4: it may be worth pointing out here the wild-type and mutant proteins expressed at similar levels; clearly the outcomes will depend on protein concentration in the cell. I believe the supplementary figure shows this to a large extent.

Indeed, our supplementary figure shows that the WT and mutant protein express to comparable levels. We have now pointed this out in the text.

11. Page 4 (Results: The adaptor) paragraph 1 line 4: "CARD domain" would stand for "caspase activation and recruitment domain domain". Please check throughout (including Supplementary Material).

We have fixed this mistake.

12. Page 4 (Results: The adaptor) paragraph 1 line 9: "expressed over a range of concentrations in cells" – this would imply the authors controlled expression – please rephrase to explain what exactly was done.

We have now rephrased this sentence to indicate that the range of expression results from the use of a genetic construct with cell-to-cell variation in copy number.

13. Page 5 (Results: The adaptor) paragraph 2 line 3 and throughout (including Supplementary Material): please use the Greek letter rather that "u" for micro.

We have now fixed this mistake.

14. Page 5 (Results: The adaptor) paragraph 3: this analysis is rather simplistic, it is not just the RMSD value, it is the nature of conformational change that is important? Please elaborate, I would think the papers presenting structural work have already discussed this to some extent?

The reviewer is correct; it is the nature of the conformational change that is most important. We are unsure how to accurately estimate the energy barrier separating the two conformations for each protein. However, we have now undertaken a collaboration to attempt to do so via FAST molecular simulations (Zimmerman and Bowman 2015). In lieu of the results of these ongoing studies, we have modified the text to acknowledge that RMSD does not necessarily relate to nucleation barriers.

Maxwell I. Zimmerman and Gregory R. Bowman. Journal of Chemical Theory and Computation, 2015, 11 (12), 5747-5757 DOI: 10.1021/acs.jctc.5b00737

15. Page 5 (Results: The adaptor) paragraph 4 line 5 and further in this section: some symbol(s) do not show in the pdf – before "(δ)", next page line 3-5 after "higher" and "both".

We have fixed this issue that resulted from exporting to a PDF file from our text editor.

16. Page 6 (Results: The adaptor) paragraph 4: interface IIa and IIIb are not introduced, and there is not even any reference provided here.

We have now added a reference for these mutations and elaborated on the interfaces IIa and IIIb.

17. Page 6 (Results: Pathogenic) paragraph 1 line 12: "FL" is not introduced.

We have now fixed this mistake.

18. Page 8 (Results: Pathogenic) paragraph 7: the text "absent the pathogenic mutations" is missing something.

We have now reworded this section.

19. Page 10 (Results: BCL10) paragraph 3: why does CARD9 CARD clustering peak and then disassemble (I guess "clustering" doesn't disassemble, please rewrite as well).

We have now fixed this mistake.

20. Page 11 (Results: MALT1) paragraph 1: I presume dimerization doesn't achieve the same level of proximity as higher-order multimerization?

Our interpretation here is that for MALT1, activation requires close proximity of more than two molecules. Although our dimerization module did not activate the caspase-like domain of MALT1, we know that it achieves close enough proximity to activate the caspase domain of CASP8. Hence we believe the MALT1 mechanism has a stoichiometry requirement in addition to a proximity requirement. This is, of course, consistent with the fact that activation normally occurs in the context of polymers rather than dimers.

21. Page 11 (Results: Ancient) paragraph 1 line 4: is this AlphaFold2?

That is correct, we used AlphaFold2. We have added that detail.

22. Page 12 (Discussion) paragraph 4: not sure if "molecular examples of evolutionary spandrels" will be clear to most readers.

We have now explained what evolutionary spandrels are, and elaborated on the relationship to our findings.

23. Page 14 (Materials: Plasmid) line 2 and throughout: "Golden Gate" is usually capitalized. Similar for "Gibson" further in the paragraph. The English in this paragraph is not up to standard in general; for example "Then placing…" is not a complete sentence, and a number of sentences ending with "via gibson" need to be rewritten.

We have now rewritten this paragraph.

24. Page 16 (Materials: Cell) line 4 and throughout: "2" in "CO2" should be subscripted.

This is now fixed.

25. Page 16 (Materials: Transient) line 6 and throughout (including Supplementary Material): please use a space between number and unit ("35 mm").

This is now fixed.

26. Page 16 (Materials: Generation) line 4 and throughout: to distinguish from "gram", please italicize "g" and/or use "x g".

We have now fixed this.

27. Page 17 (Materials: Yeast) line 3: please specify which table is "table X".

We have now fixed this mistake.

28. Page 17 (Materials: Mammalian) line 1: please provide full reference. Same next paragraph line 2.

We have now fixed this.

29. Page 17 (Materials: DAmFRET) line 3: "SSC" and "FSC" are not defined.

We have now fixed this.

30. Page 18 (Materials: Fluorescence) line 10: "Coefficient" does not have to be capitalized. It does not have to be defined again in the next paragraph.

We have now fixed this.

31. Page 19 (Materials: Optogenetic) line 1: "performed" rather than "made"?

We have now fixed this.

32. Page 19 (Materials: Protein) line 12: the Compass software doesn't have a reference?

We have now added the reference to the software.

33. References: please make format consistent: articles titles in sentence or title case.

We have now formatted all references to be consistent.

34. Legend to Figure 1: I suggest "Schematic diagram"; and "h" rather than "hrs"; please check throughout (including Supplementary Material).

We agree with this suggestion.

35. Legend to Figure S1: is "TNF-a" supposed to be "TNF-α"?

We have fixed this.

36. Legend to Figure S7: please capitalize "Figure 2H".

We have fixed this.

37. Legend to Figure S10F: please move "Dox" behind the concentration.

We have fixed this.

38. Figure S14B: the colours in the superposition make it difficult to see the differences.

We have used a different color now.

39. Legend to Figure S14: I suggest "structure…predicted by AlphaFold" (2?) and include the reference.

We agree with this suggestion.

Reviewer #4 (Significance (Required)):As argued above, the significance of this paper is that it tests directly important hypotheses proposed or assumed previously, and does so at a technically high standard. No published report has done so to a similar extent.The paper should be of interest to a broad audience from cell biologists and immunologists to biochemists, biophysicists and structural biologists.My expertise is in structural biology or systems similar to the one studied here.